# A meta-analysis of genetic effects associated with neurodevelopmental disorders and co-occurring conditions

A systematic understanding of the aetiology of neurodevelopmental disorders (NDDs) and their co-occurrence with other conditions during childhood and adolescence remains incomplete. In the current meta-analysis, we synthesized the literature on (1) the contribution of genetic and environmental factors to NDDs, (2) the genetic and environmental overlap between different NDDs, and (3) the co-occurrence between NDDs and disruptive, impulse control and conduct disorders (DICCs). Searches were conducted across three platforms: Web of Science, Ovid Medline and Ovid Embase. Studies were included only if 75% or more of the sample consisted of children and/or adolescents and the studies had measured the aetiology of NDDs and DICCs using single-generation family designs or genomic methods. Studies that had selected participants on the basis of unrelated diagnoses or injuries were excluded. We performed multilevel, random-effects meta-analyses on 296 independent studies, including over four million (partly overlapping) individuals. We further explored developmental trajectories and the moderating roles of gender, measurement, geography and ancestry. We found all NDDs to be substantially heritable (family-based heritability, 0.66 (s.e. = 0.03); SNP heritability, 0.19 (s.e. = 0.03)). Meta-analytic genetic correlations between NDDs were moderate (grand family-based genetic correlation, 0.47 (s.e. = 0.06); grand SNP-based genetic correlation, 0.39 (s.e. = 0.19)) but differed substantially between pairs of disorders. The genetic overlap between NDDs and DICCs was strong (grand family-based genetic correlation, 0.62 (s.e. = 0.20)). While our work provides evidence to inform and potentially guide clinical and educational diagnostic procedures and practice, it also highlights the imbalance in the research effort that has characterized developmental genetics research.

Neurodevelopmental disorders (NDDs) are complex health concerns, starting from childhood[1]. NDDs affect around 15% of children and adolescents worldwide and lead to impaired cognition, communication, adaptive behaviour and psychomotor skills[2]. The fifth edition of the

Diagnostic and Statistical Manual of Mental Disorders (DSM-5) categorizes the following seven disorders under NDDs: intellectual disabilities, communication disorders, autism spectrum disorder (ASD), attention-deficit/hyperactivity disorder (ADHD), specific learning

✉e-mail: a.gidziela@qmul.ac.uk; m.malanchini@qmul.ac.uk

**Fig. 1 | Visual summary of the three core aims of the current meta-analysis.** Aim 1 (orange and light blue): estimate family-based genetic ($h^2$), shared environmental ($c^2$) and non-shared environmental ($e^2$) influences as well as SNP heritability ($h^2_{SNP}$) for all NDDs identified by the DSM-5. Aim 2 (orange and red): provide grand estimates of family-based genetic ($r_A$), shared environmental ($r_C$) and non-shared environmental ($r_E$) correlations and SNP-based genetic correlations ($r_{G_{SNP}}$) between different NDDs. Aim 3 (navy blue and red): provide grand estimates of $r_A$, $r_C$, $r_E$ and $r_{G_{SNP}}$ between NDDs and DICCs. The results for $c^2$, $e^2$, $r_C$ and $r_E$ are presented in Supplementary Note 1.

disorders, motor disorders and other NDDs[3]. NDDs often have lifelong trajectories: they can manifest before 12 months of age[4] and can be diagnosed before children enter primary education[3,5].

While some NDDs (for example, ASD and ADHD) may persist throughout adolescence and adulthood[6,7], others are more likely to alleviate as children get older (for example, tic disorder[8] and communication disorders[9]). Nevertheless, all NDDs can lead to social and behavioural difficulties and reduced independence over the lifespan[6,7,10]. For instance, ADHD in childhood has been associated with an increased risk of educational and occupational problems, risk-taking, and mood disorders in adulthood[11]; and an ASD diagnosis in childhood has been associated with increased occupational difficulties and a greater risk of psychopathologies in adulthood[12,13]. The difficulties are often more salient for children diagnosed with more than one NDD[14].

A systematic understanding of the aetiology of NDDs remains incomplete. A disproportionate number of studies and systematic reviews have focused on ASD and ADHD, pointing to their substantial heritability—the extent to which observed individual differences are accounted for by underlying genetic differences. A meta-analysis of seven twin studies of clinically diagnosed ASD in child and adolescent samples yielded a grand heritability ($h^2$) estimate of 0.74 (ref. [15]). Similarly sizeable heritability estimates have also been obtained from twin studies of ADHD in childhood and adolescence[16]. Heritability estimates were found to differ across the two major components of ADHD, with genetic factors playing a more substantial role in the aetiology of hyperactivity ($h^2 = 0.71$) than that of inattention ($h^2 = 0.56$)[17]. However, other NDDs, despite showing similar prevalence rates and severity as ASD and ADHD, are less well understood and studied[18].

In line with what has been observed for all complex traits, heritability estimates for ASD and ADHD obtained from DNA data are lower than those obtained from twin and family designs[19]. Single nucleotide polymorphism (SNP) heritability can be calculated using large samples of individual-level genotype data[20] or summary statistics from genome-wide association studies[21], hypothesis-free studies aimed at discovering associations between genetic variation across the genome and individual differences in traits and disorders. The two largest

studies to date that have estimated the SNP heritability of ASD and ADHD report estimates of $h^2 = 0.12$ for ASD[22] and $h^2 = 0.22$ for ADHD[23].

It is now well established that NDDs often co-occur with one another (a phenomenon known as homotypic co-occurrence), and this points to a shared underlying liability between conditions[24,25]. Even in this instance, most studies have focused on examining the genetic correlations—the degree to which the same genetic variants contribute to the observed covariation between pairs of traits or disorders[26]—between ASD and ADHD, resulting in a meta-analytic genetic correlation of 0.59 (ref. [27]) across twin and family studies, and a SNP-based genetic correlation of 0.35 (ref. [28]). Aetiological sources of co-occurrence between all other NDDs have not been meta-analysed, but individual studies point to a moderate to strong shared liability between ASD/ADHD and other NDDs[29–33].

Another category of disorders that begin in and progress through childhood and adolescence are disruptive, impulse control and conduct disorders (DICCs), which the DSM-5 describes as disorders that share the underlying features of impulsive behaviour, aggressiveness and pathological rule breaking[3]. The DSM-5 identifies eight main DICC categories: oppositional defiant disorder, intermittent explosive disorder, conduct disorder, antisocial personality disorder, pyromania, kleptomania, other specified DICC disorder and unspecified DICC disorders[3] (Fig. 1). Similar to NDDs, DICCs have been linked to impaired social, emotional and educational outcomes[34–37].

The developmental nature of DICCs makes them an ideal primary target for the investigation of how NDDs co-occur with other disorders (that is, heterotypic co-occurrence) during childhood and adolescence. However, the distinction between NDDs and DICCs in the published literature is often blurred, particularly for disorders that include clinical features that overlap across NDD and DICC categories, such as ADHD. The most investigated examples of symptom overlap between NDDs and DICCs involve ADHD and conduct disorder[38,39], and ADHD and oppositional defiant disorder[40]. Studies highlight how these disorders are characterized by disturbances in emotion regulation, attention problems, cognitive inflexibility and impaired inhibition[39,41,42]. A shared symptomatology has also been observed between ASD and antisocial behaviour/personality disorder (which we refer to as conduct disorder

in the current work since antisocial personality disorder describes adult diagnoses[3,43,44]. However, studies on the association between NDDs and DICCs are characterized by a great deal of heterogeneity and inconsistencies across co-occurring conditions[45,46].

With three core aims (Fig. 1), the current meta-analysis bridges gaps in our knowledge of the aetiology of NDDs and their co-occurrence with other developmental conditions in childhood and adolescence. First, we meta-analysed studies on the relative contributions of genetic and environmental influences to all NDD categories described in the DSM-5. Second, we meta-analysed estimates for the genetic and environmental overlap between different NDDs (homotypic co-occurrences). Third, given their developmental onset and progression and partly shared symptomatology, we examined the aetiology of the co-occurrence between NDDs and DICCs (heterotypic co-occurrences). In addition to addressing each disorder individually, we took a transdiagnostic approach by combining data across NDDs and including categorical (that is, the presence or absence of a disorder) and quantitative (that is, continuously measured symptoms) measures. Clarifying the genetic and environmental aetiology of all NDDs and their homotypic and heterotypic co-occurrences will advance our knowledge of how developmental disorders cluster together, which could inform educational and clinical practice[47].

## Results

This section presents meta-analytic findings on genetic influences on NDDs and on their genetic overlap with other NDDs and DICCs. Meta-analytic estimates for shared and non-shared environmental factors and their overlap are presented in Supplementary Note 1. The results for all sub-categories of NDDs and DICCs are reported in Supplementary Note 2, Supplementary Figs. 2 and 3, and Supplementary Tables 2, 4 and 6.

### Searches and screening

Studies for this meta-analysis were selected during three screening stages including title and abstract screening, full text screening, and reference list screening (see Methods for a detailed description). This selection process resulted in a total of 296 studies (292 family-based and 34 SNP-based studies) included in the current meta-analysis (Fig. 2). The numbers of family-based and SNP-based studies do not add up because some studies provided both family-based and SNP-based estimates. These studies were counted only once towards the grand total but were included separately in family-based and SNP-based categories.

### Heritability of NDDs

Our first aim was to obtain reliable estimates of the contributions of genetic factors to individual differences in all NDDs. We considered two broad categories of methods that allow for the estimation of heritability: family-based designs including related individuals (such as sibling comparisons and twin studies) and SNP heritability[48] (Methods). Given the substantial differences in methodology and outcomes, the findings across these two broad categories were meta-analysed separately.

**Family-based heritability.** We identified a total of 236 family-based studies, comprising 2,792,511 partly overlapping individuals, that investigated the proportion of variance in NDDs that is accounted for by genetic factors. Out of the total, 121 studies ($N$ = 682,340) investigated ADHD, 89 studies ($N$ = 360,920) investigated specific learning disorders, 36 studies ($N$ = 1,821,970) investigated ASD, 23 studies ($N$ = 130,757) investigated communication disorders, 6 studies ($N$ = 52,278) investigated motor disorders and 2 studies ($N$ = 9,036) investigated intellectual disabilities. Across all NDDs and 236 studies, the grand $h^2$ estimate was 0.66 (s.e. = 0.03). Grand $h^2$ estimates differed, albeit not significantly, across NDD categories, ranging from 0.86 (s.e. = 0.44) for intellectual disabilities to 0.62 (s.e. = 0.04) for specific learning disorders (Fig. 3 and Supplementary Table 1). The

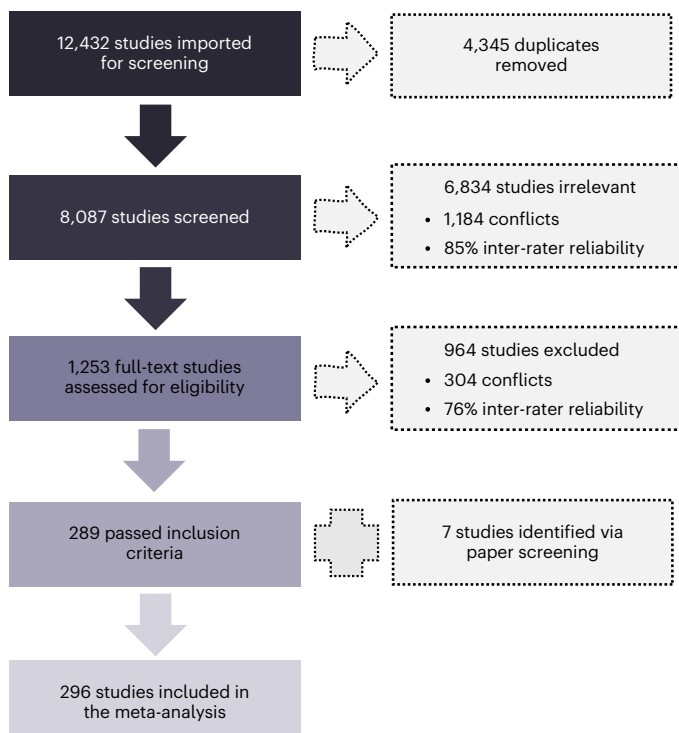

**Fig. 2 | Diagram of searches and screening.** Overview of the screening and selection process across primary and secondary searches, along with statistics of inter-rater reliability.

distributions of genetic influences across studies and NDDs are presented in Supplementary Fig. 1.

**SNP heritability.** Out of the total of 29 SNP-based studies, involving 893,896 partly overlapping individuals, the only disorders that were addressed by at least two independent studies[49] included ASD (15 studies; $N$ = 637,240), ADHD (14 studies; $N$ = 725,168), specific learning disorders (9 studies; $N$ = 40,637) and communication disorders (4 studies; $N$ = 14,894). SNP heritability across all NDDs was moderate (0.19, s.e. = 0.03) and ranged from 0.15 (s.e. = 0.04) for ASD to 0.30 (s.e. = 0.14) for communication disorders (Fig. 3 and Supplementary Table 1). SNP heritability estimates were not found to differ significantly across disorders, although the degree of precision in the estimates varied substantially depending on the sample size and the number of individual studies included per disorder.

### Genetic overlap between NDDs

Compared with the vast number of studies that examined the aetiology of individual differences in each NDD, only a limited body of research (37 studies; $N$ = 212,569) investigated the co-occurrence between NDDs in childhood and adolescence. In fact, for some of the disorders, we were unable to find two independent statistics[49] and therefore could not provide a meta-analytic estimate.

**Family-based genetic correlations ($r_A$).** When considering family-based designs (Methods and Supplementary Note 3), we obtained a sufficient number of studies to allow for meta-analysis for the following NDD pairs: ADHD and specific learning disorders (15 studies; $N$ = 67,039), ASD and ADHD (6 studies; $N$ = 58,518), ADHD and motor disorders (2 studies; $N$ = 8,748), communication disorders and motor disorders (2 studies; $N$ = 3,950), and communication disorders and specific learning disorders (2 studies; $N$ = 42,098). Only one study was identified for the following pairs of NDDs: ASD and communication disorders ($N$ = 12,174), ASD and specific learning disorders ($N$ = 6,858),

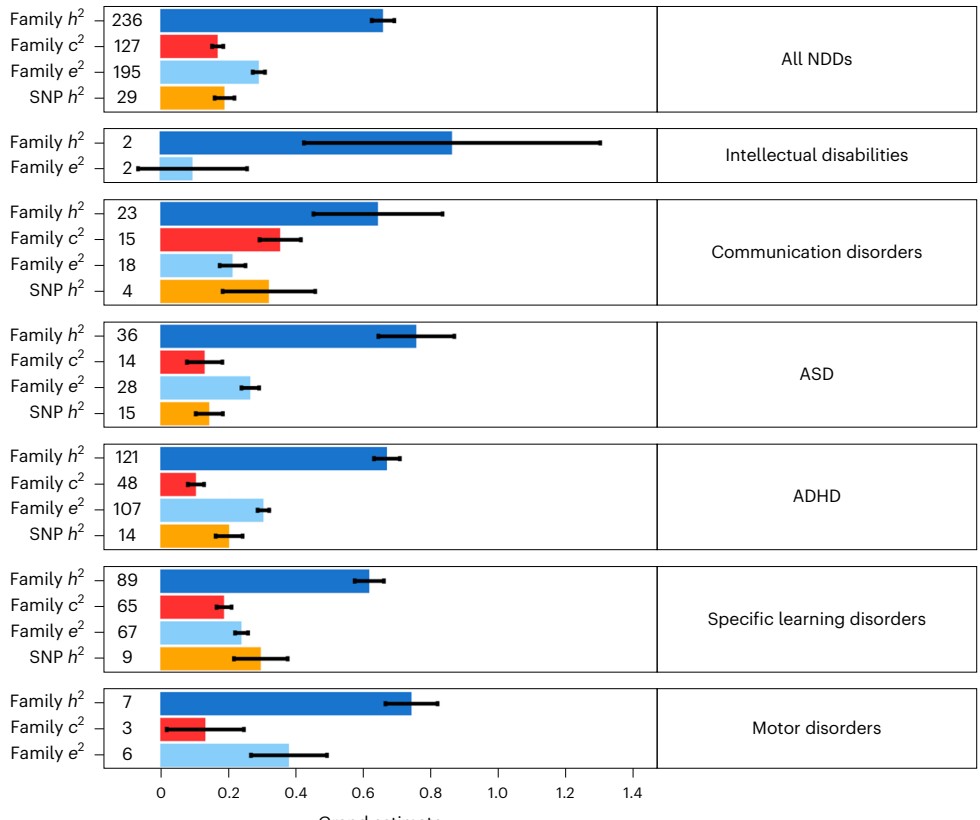

**Fig. 3 | Genetic and environmental sources of variation in NDDs.** Meta-analytic family and SNP-based heritability ($h^2$) of NDDs and shared environmental influences ($c^2$) and non-shared environmental influences ($e^2$) on variation in NDDs. The number preceding each bar on the $y$ axis denotes the number of studies identified that provided estimates for specific NDDs. The error bars signify standard errors of the grand estimates of heritability and environmental influences. The results for $c^2$ and $e^2$ are discussed in Supplementary Note 1.

ASD and motor disorders ($N = 6,858$), and specific learning disorders and motor disorders ($N = 6,858$). These studies could therefore be included only in the transdiagnostic meta-analysis, capturing the degree of genetic and environmental co-occurrence across all NDD pairs. In addition, 9 studies ($N = 46,000$) examined the co-occurrence between subtypes of specific learning disorders, such as dyslexia and dyscalculia; these studies have been included in the transdiagnostic meta-analysis, and the results of these finer-grained analyses are reported in Supplementary Note 2.

We first meta-analysed genetic correlations across all NDD categories (transdiagnostic genetic co-occurrence); this yielded a moderate grand estimate of $r_A = 0.47$ (s.e. = 0.06). When considering NDD categories separately, we found the strongest genetic overlaps between ADHD and motor disorders ($r_A = 0.90$, s.e. = 0.82) and between ASD and ADHD ($r_A = 0.67$, s.e. = 0.30), while the weakest genetic correlation was found for the association between ADHD and specific learning disorders ($r_A = 0.33$, s.e. = 0.04) and between communication disorders and motor disorders ($r_A = 0.33$, s.e. = 0.16; Fig. 4 and Supplementary Table 3). However, given the considerable differences in sample sizes used to derive genetic correlations between pairs of disorders (for example, between ASD and ADHD or between communication disorders and motor disorders), the strength of these correlations may be difficult to compare. Low correlations could also reflect low power to detect the true overlap.

**SNP-based genetic correlations ($r_{G_{SNP}}$).** SNP-based designs in child and adolescent samples exclusively focused on the association between ASD and ADHD (five studies; $N = 242,543$) and between subtypes of specific learning disorders (one study; $N = 4,500$). The transdiagnostic genetic correlation obtained via meta-analysing SNP-based designs was 0.39 (s.e. = 0.19) (Supplementary Table 8), in line with the estimate obtained from family-based designs. A grand genetic correlation of 0.20 (s.e. = 0.14) was found for the co-occurrence between ADHD and ASD. The one remaining study examined the co-occurrence between dyslexia- and dyscalculia-related traits, specifically reading and mathematics abilities, which were strongly correlated ($r_{G_{SNP}} = 0.74$, s.e. = 0.17)[50].

**Genetic overlap between NDDs and DICCs**
Our third aim was to obtain meta-analytic estimates of the genetic associations between NDDs and DICCs. Our search yielded only 15 eligible family-based studies ($N = 42,718$) and no SNP-based studies. Meta-analytic genetic correlations could be calculated for only a few NDD and DICC pairs—namely, ADHD and conduct disorder (6 studies; $N = 11,308$), ADHD and oppositional defiant disorder (6 studies; $N = 10,748$) and ASD and conduct disorder (3 studies; $N = 24,564$). In addition, we identified one study ($N = 360$) that examined the co-occurrence between specific learning disorders and disruptive behaviour, finding a weak negative genetic correlation ($r_A = -0.14$, s.e. = 0.06)[51].

**Family-based genetic correlations ($r_A$).** Across all co-occurrences between NDDs and DICCs (15 studies), the grand genetic correlation was 0.62 (s.e. = 0.20). Similarly strong genetic correlations were observed between ADHD and conduct disorder (6 studies) and between ADHD and oppositional defiant disorder (6 studies): $r_A = 0.66$ (s.e. = 0.36) and $r_A = 0.66$ (s.e. = 0.18), respectively—a similar level of aetiological overlap to that observed between strongly genetically correlated NDDs such as

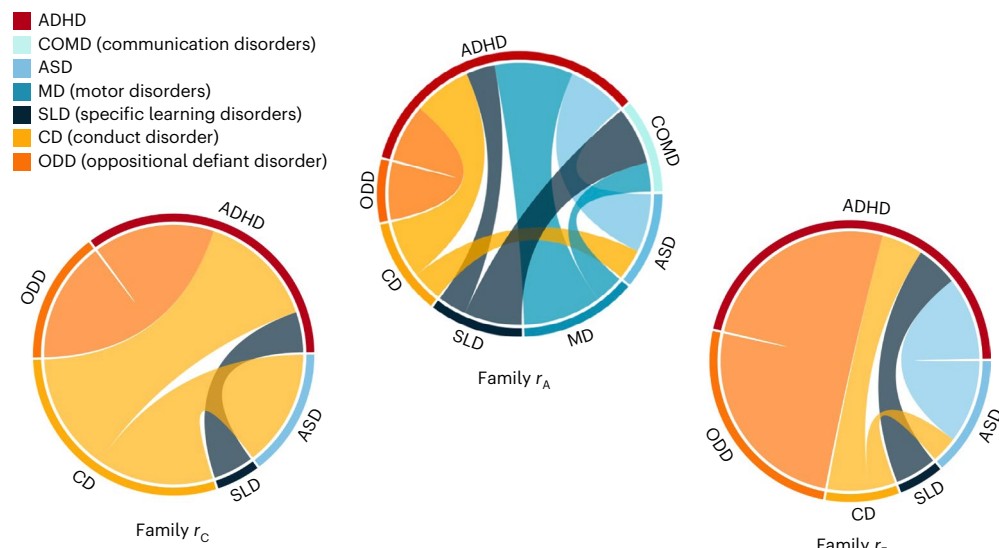

**Fig. 4 | Genetic and environmental correlations between NDDs and DICCs.**
Strength of the meta-analytic genetic ($r_A$), shared environmental ($r_C$) and non-shared environmental ($r_E$) correlations between NDDs and their homotypic (other NDDs) and heterotypic (DICCs) co-occurrences. The outer layer of each circle shows all the different NDDs and DICCs for which meta-analytic correlation estimates could be computed. Each coloured connector path indicates the strength of association between disorders; the thicker the connector path, the stronger the correlation between the two disorders. The results for family $r_C$ and $r_E$ are presented in Supplementary Note 1.

ADHD and ASD (Supplementary Table 5). In contrast, the genetic overlap between ASD and conduct disorder (3 studies) was much weaker, with a meta-analytic genetic correlation of 0.35 (s.e. = 0.10; Fig. 4). The similar extent of genetic overlap between ADHD and conduct disorder or oppositional defiant disorder and between ADHD and ASD may not be free from biases introduced by an unbalanced sample size used to derive these meta-analytic estimates. In addition, large meta-analytic standard errors make assessing the significance of differences between the estimates difficult.

## Sex differences
Some NDDs do not affect males and females equally—for instance, males are four times more likely to be diagnosed with ASD[52,53] and twice as likely to be diagnosed with ADHD[54]. Studies have suggested that these differences in prevalence may be caused by quantitative genetic sex differences, differences in the degree to which genes influence variation in NDDs in males versus females[55]. To provide an overview of sex differences in NDDs, we conducted separate meta-analyses including all studies that had reported sex-specific estimates.

**Family-based heritability.** We identified 68 family-based studies that investigated the genetic aetiology of individual differences in NDDs in male samples and 67 studies that reported estimates for female samples. Of all the studies involving sex-stratified samples, 38 studies focused on ADHD, 21 focused on ASD, 8 focused on specific learning disorders, 4 focused on communication disorders and 2 focused on motor disorders. Across all NDDs, family-based heritability was not significantly different between males and females ($h^2 = 0.65$, s.e. = 0.06 in males; $h^2 = 0.67$, s.e. = 0.06 in females). The distributions of sex-specific family-based variance components for all NDDs (except motor disorders, for which a sufficient number of studies (>1) was not identified) are presented in Fig. 5 and Supplementary Table 16.

**SNP heritability.** Marked differences in SNP heritability were observed between males and females across all NDDs (0.19, s.e. = 0.07 for males; 0.09, s.e. = 0.10 for females). However, these estimates were based on the only two studies to date that had calculated the SNP heritability of ASD and ADHD separately by sex (Supplementary Table 16).

**Sex differences in genetic overlap between NDDs.** We identified only four family-based studies that examined homotypic co-occurrences of NDDs in males and only two studies in females. Half of these studies considered the overlap between ASD and ADHD. The other half considered the co-occurrence between ASD and communication disorders (one study in both males and females) and between developmental coordination disorder and tic disorder, two subtypes of motor disorder (one study in males only). The grand family-based genetic correlation across all NDDs was estimated at 0.86 (s.e. = 0.58) for males and 0.25 (s.e. = 0.36) for females (Supplementary Table 17).

Sex-specific grand estimates of family-based genetic correlations between specific disorders could not be calculated due to the limited number of available studies. The only exception was the co-occurrence between ASD and ADHD in males, where two studies were identified ($r_A = 0.79$, s.e. = 0.42) (Supplementary Table 17). SNP-based genetic correlations between NDDs could not be calculated for males and females separately due to a lack of studies that examined these associations separately by sex in samples of children and adolescents.

**Sex differences in genetic overlap between NDDs and DICCs.** Sources of co-occurrence between NDDs and DICCs could be estimated only between ADHD and conduct disorder and only in females. Of the only two studies that examined the sex-specific co-occurrence between ADHD and conduct disorders, one used a female-only sample. Hence, we could only meta-analyse the co-occurrence between ADHD and conduct disorder in females. We found a meta-analytic genetic correlation of 0.75 (s.e. = 0.58) (Supplementary Table 18).

## Developmental trajectories
We investigated developmental change and continuity in the relative contributions of genetic factors to NDDs by examining age-related differences in their aetiology and sources of their homotypic and heterotypic co-occurrences. We distinguished among the three following developmental stages: childhood (4–7 years), middle childhood (8–10 years) and adolescence (11–24 years). We grouped estimates in any of those three categories or across multiple categories—that is, childhood and middle childhood (4–10 years), middle childhood and adolescence (8–24 years), and childhood and adolescence (4–24 years).

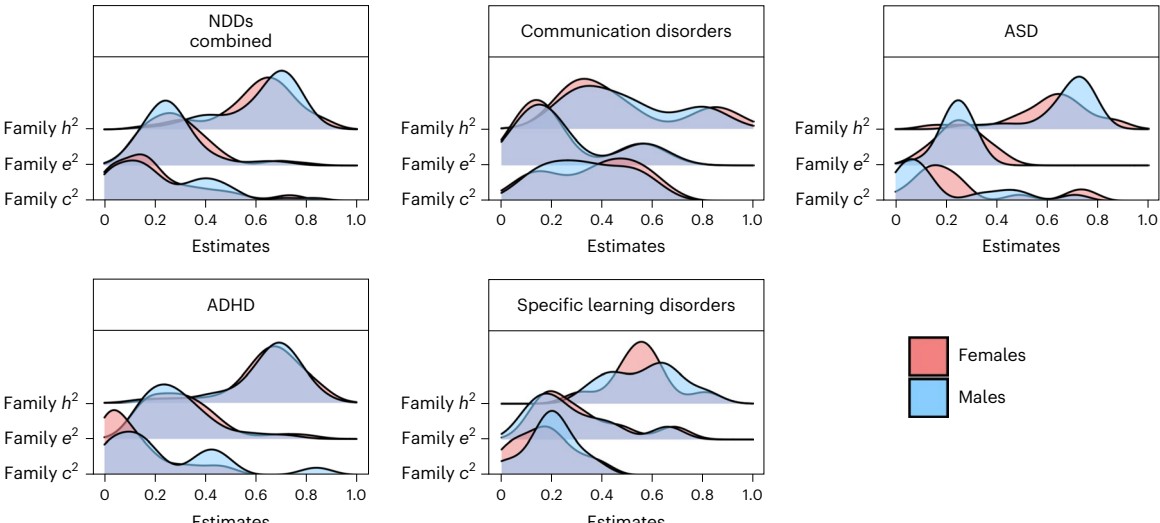

**Fig. 5 | Sex differences.** Distributions of the sex-specific meta-analytic estimates for the heritability ($h^2$) of NDDs and environmental contributions to NDDs. The top left panel shows the distributions of sex-specific estimates for the transdiagnostic meta-analysis; the remaining panels show the same estimates for specific NDDs for which a sufficient number of studies (>2) reporting sex-specific estimates was identified. The results for sex-specific $c^2$, $e^2$, $r_C$ and $r_E$ estimates are presented in Supplementary Note 1.

**Family-based heritability.** Across all NDDs, 54 family-based studies reported estimates in childhood (4–7 years), 54 studies reported estimates in middle childhood (8–10 years) and 79 studies reported estimates in adolescence (11–24 years). The remaining studies involved populations whose age range spanned across categories— that is, childhood and middle childhood (4–10 years; 14 studies), middle childhood and adolescence (8–24 years; 50 studies), and childhood and adolescence (4–24 years; 40 studies). We investigated age-related differences in heritability including all NDD categories (Fig. 6a), except motor disorders, for which we did not identify enough studies (>1) per age category. All estimates with standard errors, including those for age cross-categories, are presented in Supplementary Table 19.

Across all NDDs, grand heritability remained relatively stable developmentally, with the estimate of 0.63 (s.e. = 0.03) in childhood, a slight increase in middle childhood (0.68, s.e. = 0.04) and a subsequent drop back to 0.62 (s.e. = 0.08) in adolescence. This trend was consistent for some specific disorders (for example, ASD and ADHD) but not for others (for example, communication disorders and specific learning disorders), for which genetic influences decreased developmentally (Fig. 6a and Supplementary Table 19).

**SNP heritability.** Of a total of 29 SNP-based studies that were identified, 13 included adolescent samples, 7 included samples in middle childhood and 6 included samples in childhood, while 11 studies reported estimates across childhood and adolescence. SNP heritability was stable developmentally across NDDs, and the developmental trajectory mirrored that of family-based heritability (SNP $h^2$ = 0.24, s.e. = 0.11 in childhood; SNP $h^2$ = 0.26, s.e. = 0.08 in middle childhood; SNP $h^2$ = 0.23, s.e. = 0.07 in adolescence) (Fig. 6b and Supplementary Table 19). For ASD, ADHD and specific learning disorders (the specific NDDs for which grand estimates could be calculated), the developmental trends were consistent with those observed for family-based heritability (Fig. 6b and Supplementary Table 19).

**Developmental trajectories in genetic overlap between NDDs.** Overall, we could not explore developmental trends in genetic correlations using either method due to a lack of available studies; the only exceptions were grand estimates for adolescence and across age categories (Supplementary Tables 20 and 21). Genetic correlations obtained for adolescent samples only were in line with those obtained for the total sample (for example, when considering the co-occurrence between ASD and ADHD, the genetic correlation was 0.66 (0.49) in adolescent samples and 0.67 (0.30) across all age categories).

**Categorical versus continuous measurement**
Although we meta-analysed categorical (binary phenotypes, such as clinical diagnoses and cut-offs) and quantitative (sub-threshold symptom counts or test/questionnaire scores) measures together, we also report separate grand estimates for both measurement types. Across all NDDs, categorical measures were observed to yield significantly higher family-based heritability estimates than continuous phenotypes (0.77 (s.e. = 0.07) versus 0.64 (s.e. = 0.03)). However, the opposite was found for SNP-based heritability (0.17 (s.e. = 0.03) for categorical measures versus 0.25 (s.e. = 0.06) for quantitative assessments). Differences in sources of variation in specific NDDs as well as specific homotypic and heterotypic co-occurrences are presented in Supplementary Note 4, Supplementary Fig. 26 and Supplementary Tables 28–30.

**Geography and ancestry**
Research into the genetic aetiology of NDDs and of their homotypic and heterotypic co-occurrences is largely limited to Western countries, even though, according to the Global Burden of Disease study[56], the prevalence of diagnosed NDDs is not uniform across the globe. Furthermore, individuals of European ancestry represent 16% of the global population but 80% of participants in genomic (that is, DNA-based) research[57]. This Eurocentric bias[58] has created a major gap in our knowledge of the genetic aetiology of NDDs and their co-occurrences in non-White populations. In the following section, we provide an overview of how behaviour genetics research into NDDs is distributed across countries and continents and how the estimates differ as a function of geographical location. Supplementary Note 5, Supplementary Fig. 27 and Supplementary Tables 25–27 contain meta-analytic results of how heritability and genetic correlations differ at different levels of sample ancestral diversity. We created a moderator with four levels of percentage of European-ancestry participants in samples: less than 50%, more than 50% but less than 75%, more than 75% but less than 100% and 100%.

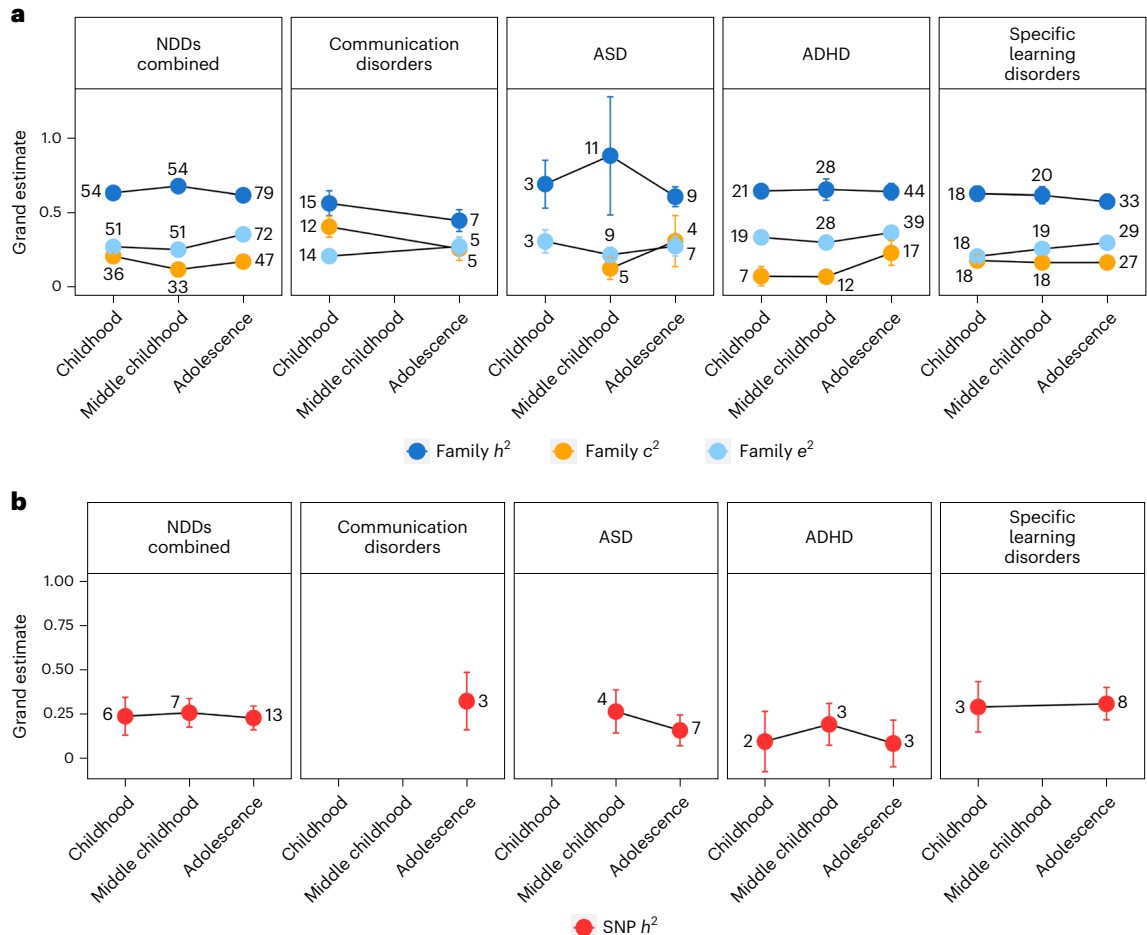

**Fig. 6 | Developmental trajectories. a,b,** Age-related differences in family-based heritability ($h^2$) and shared ($c^2$) and non-shared ($e^2$) environmental influences on NDDs (**a**) and SNP heritability (**b**). Developmental stages include childhood (4–7 years), middle childhood (8–10 years) and adolescence (11–24 years). The error bars represent standard errors of grand estimates of heritability and environmental influences. The number located near each point estimate denotes the number of studies identified that provided estimates for specific developmental stages. For intellectual disabilities and motor disorders, we could not identify a sufficient number of studies (>1) reporting age-dependent estimates, and we were consequently unable to derive meta-analytic estimates. The results for age-stratified $c^2$, $e^2$, $r_C$ and $r_E$ are reported in Supplementary Note 1.

**Family-based heritability.** Of the 236 studies investigating sources of individual differences in NDDs, 41% (96 studies) involved samples and cohorts based in the United Kingdom, 77 studies involved samples based in the United States, 24 studies involved Swedish samples, 19 studies involved Dutch samples, 11 studies involved Australian samples, 7 studies involved Canadian samples, 4 studies involved samples from China and 2 studies involved samples from Norway. Other countries that contributed to the total grand estimate but did not have enough estimates for separate meta-analysis (that is, only one study was found from each country) included Finland, Japan, South Korea and Italy. Estimates differed significantly across countries. Considering all NDDs, the highest meta-analytic family-based heritability was estimated for Australian and Swedish samples (0.76 (s.e. = 0.17) and 0.74 (s.e. = 0.05), respectively), while the lowest was obtained for Canadian cohorts (0.43 (s.e. = 0.09)) (Fig. 7a and Supplementary Table 22).

Specific NDDs were investigated with different frequencies across countries: the aetiology of intellectual disabilities was exclusively investigated in Swedish cohorts (2 of 2 studies), and most studies addressing sources of variance in motor disorders also came from Sweden (4 of a total of 7 studies). Communication disorders were mostly researched in the United Kingdom (17 of a total of 23 studies), as were ASD (20 of 36 studies) and ADHD (42 of 121 studies). In contrast, 47 of a total of 89

studies investigating specific learning disorders were carried out in the United States.

**SNP heritability.** Studies exploring the SNP heritability of NDDs focused entirely on European cohorts and were primarily conducted in the United Kingdom and the Netherlands (14 and 3 of 29 SNP-based studies in total) (Supplementary Table 22).

**Geography- and ancestry-related differences in the genetic overlap between NDDs.** Sources of homotypic co-occurrence with NDDs were investigated in 37 independent family-based studies, of which the majority were conducted in the United Kingdom (49%) or the United States (30%). The highest genetic correlation across all co-occurrences was estimated in Swedish cohorts (0.80, s.e. = 0.26 across three studies), while the lowest grand genetic overlap was estimated in Canadian samples (−0.44, s.e. = 0.24 across only two studies that investigated the association between ADHD and specific learning disorders; Fig. 7b and Supplementary Table 23).

The genetic aetiology of the co-occurrence between ASD and ADHD during childhood and adolescence was exclusively researched in the United Kingdom and Sweden (three of a total of six studies each). The co-occurrence between ADHD and motor disorders was explored by only two studies, one conducted in Sweden and the other

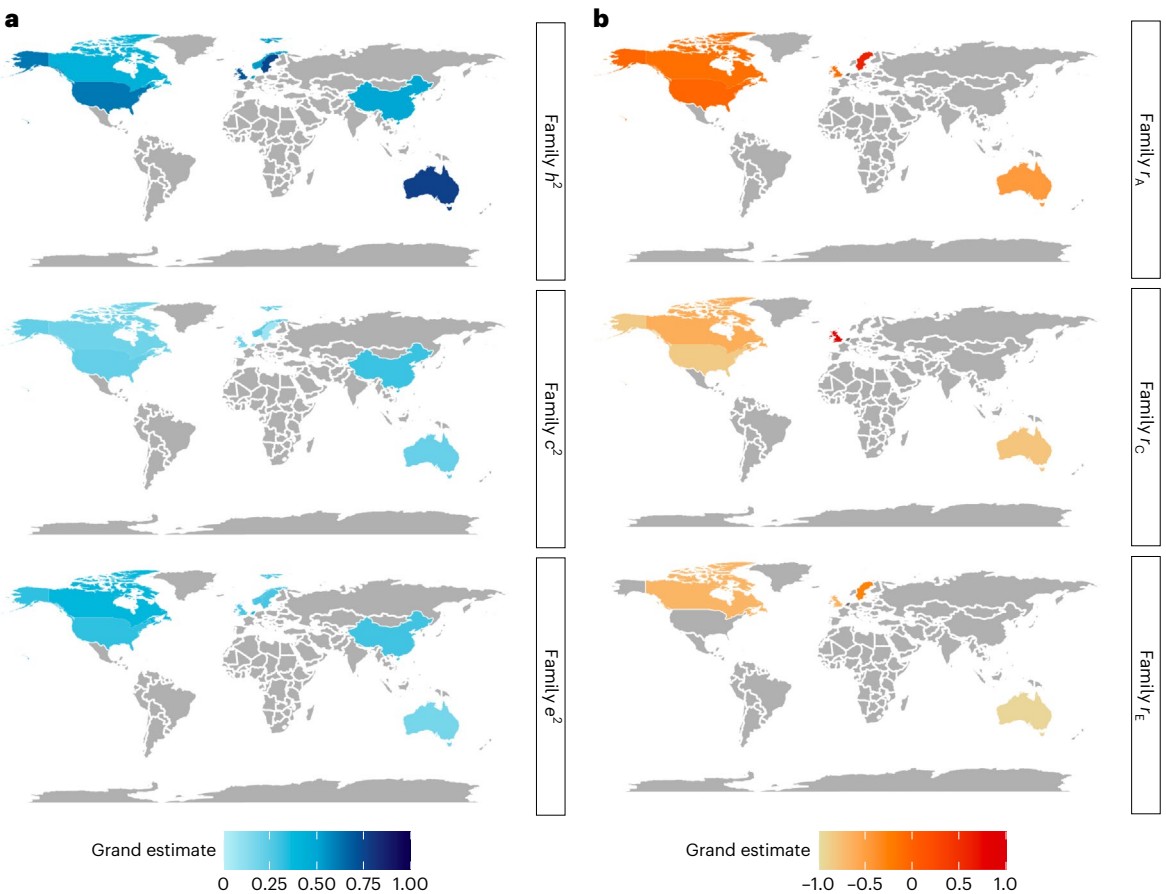

**Fig. 7 | Geographical differences. a**, Differences in family-based heritability ($h^2$) and shared environmental ($c^2$) and non-shared environmental ($e^2$) influences across all NDDs. **b**, Geographical differences in the genetic ($r_A$), shared environmental ($r_C$) and non-shared environmental ($r_E$) overlap between NDDs.

The areas shaded in grey are regions for which not enough relevant studies were identified (<2 studies). Geographical differences in $r_A$, $r_C$ and $r_E$ between NDDs and DICCs are presented in Supplementary Fig. 28. The results for $c^2$ and $e^2$ as well as $r_C$ and $r_E$ are discussed in Supplementary Note 1.

in Australia. Most studies examining the genetic overlap between ADHD and specific learning disorders came from the United States (8 of a total of 18 studies), whereas the overlap between communication disorders and motor disorders was addressed by only two studies conducted in the United Kingdom and Japan.

SNP-based studies (six in total) addressing the co-occurrence between NDDs were exclusively conducted in combined samples from the United Kingdom and Denmark (Supplementary Table 23).

**Geography- and ancestry-related differences in the genetic overlap between NDDs and DICCs.** A total of 15 family-based studies addressing the co-occurrence between NDDs and DICCs were identified, 40% of which were conducted in the United Kingdom, 20% in the United States and 20% in Sweden. The studies yielded consistently strong estimates of genetic correlations across the three regions: genetic correlations of 0.60 (s.e. = 0.29), 0.42 (s.e. = 0.15) and 0.68 (s.e. = 0.41), respectively (Supplementary Fig. 28 and Supplementary Table 24). The remaining 20% of the studies were conducted in Australia, Finland and South Korea but could not be meta-analysed separately as only one estimate was available for each country.

In terms of specific co-occurrences between NDDs and DICCs, half of the studies that explored genetic overlap between ADHD and conduct disorder and between ADHD and oppositional defiant disorder were conducted in the United States (three studies each). Three out of four studies examining the association between ASD and conduct disorder were conducted in the United Kingdom; the fourth was conducted in Sweden.

**Bias and heterogeneity assessment**
We applied $I^2$ statistics to assess heterogeneity in the estimates, followed by outlier and influential case identification analyses. The results of these analyses are reported in Supplementary Note 6, Supplementary Tables 7–12 and Supplementary Figs. 4–7. We applied Egger's regression and inspected funnel plots to examine the impact of publication bias on our results; the outcomes of these analyses are reported in Supplementary Note 7, Supplementary Tables 13–15 and Supplementary Figs. 8–24. The results of the risk-of-bias assessment are presented in Supplementary Fig. 25, where 93.8% of studies showed low risk of bias across the nine quality checklist items, and the remaining 6.2% showed moderate risk.

## Discussion
The findings of the present meta-analysis synthesize the current state of knowledge on NDDs and have implications that can guide future research strategies as well as clinical and educational practice. First, by providing estimates of the relative contributions of genetic factors to all NDDs, our work responds to the need of moving beyond the nearly exclusive research focus on ASD and ADHD. Second, by providing an account of the genetic overlap between NDDs, we highlight how genetic influences are implicated in the co-occurrence between multiple NDDs, identifying patterns of shared aetiological liability. Third, by synthesizing the literature on the co-occurrence between NDDs and DICCs, we highlight how disorders from these two separate groups identified by the DSM-5 share as much of their genetic aetiology as do disorders all classified as NDDs.

Our work provides meta-analytic evidence for the substantial heritability of all NDDs (our first aim), particularly when considering family-based studies, which indicated that around two-thirds of the variation in NDDs is accounted for by genetic differences between individuals (in children and adolescents). Although males are up to four times more likely to be diagnosed with ASD and ADHD than females[52–54], we showed that, when meta-analysed, the genetic effects associated with NDDs do not differ by sex. We also showed that genetic sources of variation in NDDs are remarkably stable across developmental stages, and this developmental stability was observed across all NDDs. Genetic effects were also mostly consistent when we separated studies that had considered diagnoses and clinical cut-offs from studies that had quantified NDDs as continuous traits.

Interestingly, we found that the genetic contributions to NDDs differed substantially as a function of geography. This highlights how estimates of genetic effects associated with disorders are sensitive to different environmental contexts[59,60]. Our work on geographical differences also highlights the major gap in our knowledge of the aetiology of NDDs in non-Western countries, a gap that is exceeded by the lack of ancestral diversity observed across all studies of NDDs. Importantly, the current study points to how genetic influences on NDDs are substantially reduced in more ancestrally diverse samples, again highlighting how heritability estimates are inextricably linked to our social context[61,62], in the sense that increased ancestral homogeneity in the sample probably entails increased environmental homogeneity, reducing environmental variability and inflating heritability in these populations.

The lack of diversity in genetic research remains its most striking limitation to date, particularly when considering DNA-based methods, hampering the extension of genetic findings to the entire population[63,64]. Limited research resources in under-represented populations are likely to have profound cascading effects for future advances in clinical practice, including pharmacological and behavioural treatment. Fortunately, there are major initiatives underway to re-balance these biases[65–67].

Our second aim was to provide a clear account of how close NDDs are to one another aetiologically. We found that, while meta-analytic estimates indicated moderate genetic overlap, the degree of heterogeneity in these associations across disorders was large. We found substantial genetic correlations between ASD and ADHD, between ADHD and motor disorders, and between communication disorders and specific learning disorders. In contrast, genetic overlap was only moderate between communication disorders and motor disorders and between ADHD and specific learning disorders, which is consistent with the degree of symptom resemblance across these disorders.

Although we were able to explore general patterns of variation and co-occurrence, the aetiology of specific NDDs and of their associations could not be comprehensively characterized. The research gaps that we identified highlight an imbalance in focus across NDDs in developmental behaviour genetics research. When considering our first aim, we could identify only 2 family-based studies that investigated the genetic contributions to intellectual disabilities, compared with 121 family-based and 14 SNP-based studies for ADHD, and 36 family-based and 15 SNP-based studies for ASD. This lack of research on intellectual disabilities, an NDD affecting 2.5% of children in the United Kingdom[68] (more than double the prevalence rate of ASD[69]), is reflected in and probably partly due to the lack of funding bodies devoted to researching NDDs other than ASD and ADHD, as well as a lack of publicly available data repositories and resources (for example, refs. [70–72]).

We also identified very few studies that examined the aetiology of motor disorders, another neurodevelopmental condition showing significant prevalence rates of 5–6% in school-aged children[73]. This unbalanced research focus, which extends far beyond genetically informative research to touch developmental and therapeutic research[74–77], has led to an uneven distribution of knowledge, which could lead to limited access to interventions for children with NDDs other than ASD, ADHD and dyslexia[78].

The lack of equity in focus across NDDs was pronounced in analyses addressing our third aim. Sources of co-occurrence between NDDs and DICCs could only be investigated between ADHD and conduct disorder, between ADHD and oppositional defiant disorder, and between ASD and conduct disorder. Considering that in the DSM-5 the DICCs category comprises eight distinct disruptive disorders, this highlights a major gap in our knowledge.

This meta-analysis provides a holistic view of genetic and environmental contributions to all NDDs and commonly co-occurring developmental disorders, revealing that NDDs are just as strongly genetically correlated with other NDDs as most of them are with DICCs. Our work identifies a lack of balance in research across different NDDs, which calls for future genetic research to focus on less-investigated disorders. We provide knowledge about patterns of aetiological co-occurrence between NDDs as well as between NDDs and DICCs, which we hope will inform clinical and educational diagnostics and practice, resulting, for example, in expanded diagnostic screening.

## Methods

The protocol for the current meta-analysis was registered with the international prospective register of systematic reviews (PROSPERO) and can be accessed at the following link: https://www.crd. york.ac.uk/prospero/display_record.php?RecordID=230158. This meta-analysis was conducted in line with the Preferred Reporting Items for Systematic Reviews and Meta-Analyses (PRISMA) guidelines[79]. The PRISMA 2020 Checklist and PRISMA 2020 for Abstracts Checklist[79] are included in Supplementary Notes 9 and 10. The code and master extraction tables are available at https://github.com/CoDEresearchlab/ Meta_analysis_NDDs_DICCs.

### Identification of relevant studies

A total of 296 studies were included in the meta-analysis (Fig. 2). Studies were identified during three searches: the primary search (Supplementary Fig. 29a), conducted on 20 January 2021; the secondary (confirmatory) search (Supplementary Fig. 29b), conducted on 15 April 2021; and the additional search of other relevant meta-analyses and reviews, finalized on 4 May 2021. The searches were conducted across three platforms: Web of Science, Ovid Medline and Ovid Embase. The outputs were managed with the aid of Covidence (Veritas Health Innovation), which is a web-based collaboration software platform that streamlines the production of systematic and other literature reviews (https://www.covidence.org/). An in-depth description of indexes, timespans, search strategy and key words is included in Supplementary Note 11. All studies included in the meta-analysis are listed in Supplementary Tables 31–36.

### Screening and inclusion criteria

After the initial searches were conducted and duplicate studies were removed, 8,087 studies met the criteria for the first stage of screening, which involved title and abstract scanning. All titles and abstracts were screened by two independent, blinded reviewers to ensure inter-rater agreement. Conflicts were resolved by a third independent reviewer, and inter-rater reliability was calculated as the number of agreements divided by the total number of studies screened, multiplied by 100% (Fig. 2). After this initial screening phase, 6,834 studies were excluded as they were deemed not relevant to the purpose of the current meta-analysis.

The title and abstract screening process resulted in a total of 1,253 potentially eligible studies. The full text of each study was screened by two independent, blinded reviewers. Reviewer discrepancies were identified and resolved by a third independent reviewer. The inter-rater reliability statistic was calculated (Fig. 2). This resulted in 289 eligible

articles. In addition, during full-text screening, relevant review articles, meta-analyses, editorials and conference abstracts were flagged to aid the potential discovery of further relevant studies by either screening the reference lists or contacting the authors of conference abstracts. Through this process, 7 additional studies were identified, which resulted in a total of 296 studies included in the current meta-analysis (Fig. 2). Studies were considered relevant and selected to be included at the next screening stage on the basis of the following criteria.

First, studies were included only if 75% or more of the sample consisted of children and/or adolescents. On the basis of guidelines from the World Health Organization (https://www.who.int/health-topics/adolescent-health#tab=tab_1), we defined the period from childhood to the end of adolescence as ranging from age 4, the earliest age for compulsory schooling, to age 24, the end of adolescence. Second, we included studies that had measured NDDs and DICCs considering formal clinical diagnoses, clinical cut-offs and/or quantitative measures of symptoms. Third, studies were selected only if they featured data on at least one NDD (Aim 1), at least two NDDs (Aim 2) or at least one NDD and one DICC (Aim 3).

Fourth, studies using family-based designs had to have reported at least one estimate of heritability ($h^2$), shared environmental ($c^2$) or non-shared environmental influence ($e^2$), or genetic or environmental correlations. We included only single-generation family designs—that is, studies that used twin designs[80], sibling comparisons[81] or extended twin designs[82]. We excluded multiple-generation family designs (for example, children of twins[83] and in vitro fertilization[84]) due to the potential confounding in the genetic and environmental estimates that could have resulted from including parental traits in the models decomposing the covariance between family members[85].

Fifth, studies using genomic designs were included only if they reported at least one SNP-based heritability estimate and/or a genetic correlation ($r_A$). Eligible SNP-based methods to quantify the proportion of phenotypic variance accounted for by common SNPs included genome-based restricted maximum likelihood[86], linkage-disequilibrium score regression[21] and SbayesS, which is a Bayesian approach to the analysis of genome-wide association summary data[87]. Each method is described in greater detail in Supplementary Note 12. Sixth, studies that selected participants on the basis of other diagnoses not related to NDD or DICC categories or on the basis of extreme vulnerability or environmental insult unrelated to NDDs or DICCs, such as alcohol abuse, were not included. Lastly, only studies published in English were included. Studies deemed eligible on the basis of full-text scanning were also scored in terms of their scientific quality and risk of bias by two reviewers (see the details on the quality-scoring checklist in Supplementary Note 13).

**Data extraction**
Data extraction was conducted by the primary reviewer. Issues and uncertainties were resolved through discussion with co-authors. Missing data were requested from study authors via email or ResearchGate (for details, see Supplementary Note 14). The extracted data were compiled in a table, including information on study reference, the project/cohort name, the study design (for example, classical twin study), the model reported (for example, the full ACE model; when multiple models were reported, the best-fitting model was selected for data synthesis), the overall number of participants and numbers of participants in subgroups (for example, the number of monozygotic versus dizygotic twins), the average age and age range of the sample, the cohort country or countries of origin, the participants' ancestry (defined in terms of the percentage of participants of European ancestry in the samples), the broad types of NDD and DICC included (for example, specific learning disorder), the subtypes of NDD and/or DICC included (for example, dyslexia), the specific phenotypes measured (for example, reading fluency), the measure statistics (for example, binary (diagnosis) or continuous (symptom continua)), the measure

(for example, Conners rating scale for ADHD) and rater (for example, parent reports), the covariates included in the analyses (for example, age and sex), statistics (for example, family-based heritability and SNP-based genetic correlation), and the estimated statistics and the provided index of measurement variance (for example, standard error). The master extraction tables, 'Extraction_heritability' and 'Extraction_correlations', are available at https://github.com/CoDEresearchlab/Meta_analysis_NDDs_DICCs.

Estimates of heritability and shared and non-shared environmental influences were extracted as reported by individual studies. When studies only reported twin correlations, the variance components were calculated using Falconer's formula[88], as follows:

$$h^2 = 2(r_{MZ} - r_{DZ})$$

$$c^2 = 1 - (h^2 + e^2)$$

$$e^2 = 1 - r_{MZ}$$

where $r_{MZ}$ is the monozygotic twin correlation and $r_{DZ}$ is the dizygotic twin correlation.

Genetic, shared and non-shared environmental correlations were extracted only if reported by individual studies. For studies where neither standard deviations, standard errors nor 95% confidence intervals were reported, the 95% confidence intervals were calculated using the Cir function implemented in the R package psychometric[89,90], on the basis of the sample size of the study, and subsequently converted to standard errors via dividing the difference between the upper and lower bounds by 3.92 (ref. [91]).

**Data synthesis**
Heritability and environmental influences reported by the selected studies were synthesized using a multilevel random-effects meta-analysis in metafor for R[55]. We used heritability/environmental influences and genetic/environmental correlation coefficients, along with standard errors, as the measures of effect size[27]. However, to avoid the risk of type I error introduced by the distribution characteristics of the correlation coefficient[92], we transformed all estimates using Fisher's z. The effect sizes were then weighted by their inverse variance weights so that larger samples were given more weighting, and the standard error for the common effect size resulted as a function of the allocated weights. To present the results, Fisher's z was transformed back to variance components and correlation coefficients[93]. Multilevel random-effects models enabled varying true effect sizes across studies. We introduced a two-level structure to account for nested effects underlying heterogeneity and clustering across studies (Level 1, individual clustering; Level 2, cohort clustering). Given that some NDDs have different prevalence rates in males and females[52–54], we meta-analysed studies that provided sex-specific estimates in separate models to minimize sample heterogeneity across studies, and we report separate grand estimates for combined, male-only and female-only samples.

**Data reporting**
We report transdiagnostic grand estimates across all disorders and for broad NDD categories, comprising all studies that investigated the aetiology of a disorder using diagnoses, categorical measures or quantitative measures. For example, the broad ADHD phenotype includes studies that have measured ADHD using diagnoses, clinical cut-offs and continuous measures of ADHD traits, such as checklists and questionnaires. The only exception is intellectual disability. We did not consider quantitative measures of general intelligence as indexing a continuum of intellectual disability given that intellectual disability, as described in the DSM-5, is a complex disorder, characterized by impairments not only in intellectual performance but also in adaptive

functioning and communication[3,44]. Finally, we considered specific manifestations of NDDs—for example, beyond ADHD, we also consider the hyperactive/impulsive and inattentive subtypes separately. The results for all sub-categories of NDDs and for their co-occurrence with other disorders are reported in Supplementary Note 2, Supplementary Figs. 2 and 3, and Supplementary Tables 2, 4 and 6.

### Aggregation of non-independent effects

Multilevel meta-analytic models allow us to account for non-independence of estimates derived from partly or completely overlapping samples (that is, estimates obtained from multiple studies that have used the same cohort of participants). To further account for the non-independence of sampling variance (that is, when sampling errors correlate because data from partly the same individuals are used to estimate multiple effect sizes), we also aggregated multiple estimates within each individual study (for example, estimates at multiple time points derived from the same study). Dependent effect sizes were aggregated at the level of each study using the R package Meta-Analysis with Mean Differences[90,94], applying a default correlation between estimates of 0.5. We conducted several sensitivity analyses comparing different aggregation methods—that is, aggregating at the level of the study, cohort and country, and varying the assumed correlation between dependent effect sizes (0.5, 0.3 and 0.9). The results of these additional checks are presented in Supplementary Fig. 30 and discussed in Supplementary Note 15. Since differences in aggregation strategy did not result in significant differences in meta-analytic effects, we report the results obtained when the correlation between dependent effect sizes was set to 0.5.

### Bias and heterogeneity assessment

The potential for publication bias was explored using funnel plots and Egger's linear regression[95]. The proportion of heterogeneity across estimates was estimated using the $I^2$ statistic, which calculates the fraction of variance across studies that can be attributed to heterogeneity rather than chance[96–98]. The $I^2$ statistic was computed as the proportion of the true variance of true effects to the variance of the observed effects, in line with the following formula:

$$I^2 = \frac{V_{\text{TRUE}}}{V_{\text{OBS}}}$$

where $V_{\text{TRUE}}$ is the variation of true effects and $V_{\text{OBS}}$ is the variation due to sampling error. In other words, $I^2$ can be interpreted as the dispersion of observed effects compared with the dispersion that would be predicted just from sampling error. The $I^2$ statistic also provides insight into the degree to which confidence intervals from individual studies are independent. We also conducted outlier case identification analysis, followed by re-calculation of the $I^2$ estimates after removing studies considered to be outliers[99]. Studies having a substantial impact on the grand estimates and heterogeneity were identified using influential case identification analysis[99]. Heterogeneity assessment analyses were conducted using the metafor[49], meta[100] and dmetar[101] packages in R[90].

### Moderation analyses

We tested for the effects of several moderators. The moderator terms were selected on the basis of the available data, considering the completeness of the reported moderator variables. We implemented a >50% rule of thumb—that is, if 50% or more studies reported data on the moderating variable, we included this moderator in our analyses. For example, less than 50% of studies reported the percentage of participants of Asian ancestry in the sample; hence, we did not include the percentage of Asian participants in the moderation analyses. We considered the following ten moderators: age group, design, type of model, rater, measurement, percentage of individuals who identified as White, number of covariates included in the analysis, measure adopted, country and specific

phenotype measured. Each moderator is described in greater detail in Supplementary Note 3. The moderation analyses were conducted using a two-step procedure. First, only studies that reported data on the level of the moderator were selected (for example, only studies reporting estimates for adolescents). Second, analyses stratified by levels of the moderator were run using a multilevel random-effects meta-analysis in metafor for R—for example, a grand estimate was derived for adolescents and subsequently compared with estimates for other developmental stages (that is, childhood and middle childhood) using the same procedure. We report unstratified estimates (Supplementary Tables 1, 3 and 5) and estimates stratified by the specific phenotype measured (Supplementary Tables 2, 4 and 6), age category (Supplementary Tables 19–21), country (Supplementary Tables 22–24) and ancestry (Supplementary Tables 25–27) in the main text, whereas estimates stratified by all other moderators are reported in Supplementary Tables 37–50.

### Deviations from the PROSPERO pre-registered protocol

Although we followed the preregistered plan step by step, we made some deviations from the plan on the basis of the availability of software and evidence. We describe our deviations from the preregistered protocol below.

(1) As opposed to the first (primary) literature search, which followed the procedure described in the protocol, in the second (confirmatory) literature search we included an additional set of terms to identify studies that measured specific learning disorders and communication disorders on a quantitative scale. For the details, see Supplementary Note 11.

(2) In the protocol, we indicated that study screening would be documented on an Excel spreadsheet. Instead, we used Covidence (https://www.covidence.org/), a software that automatically enables the double-blinded screening of title and abstract, as well as full-text screening and study selection, without the need for external recording of decisions.

(3) Finally, while all 296 papers were assessed for publication reporting bias (Supplementary Note 7, Supplementary Tables 13–15 and Supplementary Figs. 8–24), the first 82 papers that were extracted (27.7% of the total) were also assessed for study quality using the checklist provided by Kmet et al.[102] (Supplementary Note 13 and Supplementary Fig. 25).

### Certainty assessment

We evaluated our confidence in the body of research included in the present meta-analysis on the basis of a number of key factors: (1) the sample size of each study, (2) the consistency of findings across studies, and (3) study quality and risk of publication bias.

(1) Because differences in sample size can introduce an imbalance in the power to estimate effects reliably across studies, in our meta-analysis we weighted each estimate by the standard errors. Estimates reported by studies conducted in larger samples had smaller standard errors and were therefore given more weight than those reported by studies conducted in smaller samples.

(2) The consistency of findings across studies was assessed by visually examining forest plots. Overall, we did not find significant differences between estimates.

(3) Study quality and risk of bias were assessed in line with the framework proposed by Kmet et al.[102] (Supplementary Note 13 and Supplementary Fig. 25). We applied Egger's regression and inspected funnel plots to examine the impact of publication bias on our results; the outcomes of these analyses are reported in Supplementary Note 7, Supplementary Tables 13–15 and Supplementary Figs. 8–24.

On the basis of these criteria, we place confidence in the results of the current meta-analysis showing that (1) NDDs in childhood and adolescence are highly heritable; (2) the pattern of co-occurrence between

NDDs is complex, and while some NDDs are closely related, others show little genetic overlap; and (3) NDDs show a moderate-to-strong genetic overlap with DICCs.

## Limitations of the review process

The review process of the current meta-analysis does not come without limitations. One limitation is our sole focus on childhood and adolescence. A second limitation relates to our choice of focusing on specific co-occurring conditions, DICCs, without considering other neurological disorders that have been found to co-occur with NDDs, such as epilepsy, cerebral palsy, or sleep or psychiatric disorders. The inclusion of a wider range of co-occurring conditions could have resulted in a more detailed characterization of aetiological overlaps between NDDs and other conditions.

A third limitation is that the current meta-analysis focused only on single-generation studies—that is, twin and sibling studies—and excluded multi-generational family designs, such as children-of-twins and in-vitro-fertilization studies. Future studies focusing on multi-generational designs could provide valuable insights into the roles that parental genotypes and correlated environmental influences play in offspring's NDDs and their co-occurring conditions.

## Reporting summary

Further information on research design is available in the Nature Portfolio Reporting Summary linked to this article.

## Data availability

The data that support the findings, including the master extraction tables, are available at https://github.com/CoDEresearchlab/Meta_analysis_NDDs_DICCs.

## Code availability

The code for all analyses is available at https://github.com/CoDEresearchlab/Meta_analysis_NDDs_DICCs.

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

## Acknowledgements

This meta-analysis was funded by a starting grant awarded to M.M. by the School of Biological and Behavioural Sciences at Queen Mary University of London (grant no. QMULMAL18). A.G. is supported by a Queen Mary School of Biological and Behavioural Sciences PhD Fellowship awarded to M.M. G.M. was in receipt of a Klingenstein Third Generation Foundation fellowship (grant no. 20212999). F.P. is supported by a National Institutes of Health Subaward via the Regents of the University of California, Riverside (grant no. AG046938). K.R. is supported by a Sir Henry Wellcome Postdoctoral Fellowship. This research was funded in whole or in part by the Wellcome Trust (grant no. 213514/Z/18/Z). For the purpose of open access, the author has applied a CC BY public copyright licence to any Author Accepted Manuscript version arising from this submission. The funders had no role in study design, data collection and analysis, decision to publish or preparation of the paper.

## Author contributions

A.G., M.M. and Y.I.A. conceived and designed the study. A.G., L.Y.L., M.D., F.P. and E.D. conducted the literature search and study screening. A.G. extracted and analysed the data. A.G. and M.M. wrote the paper with helpful contributions from Y.I.A., G.M., A.G.A., J.A.-B., F.P., A.R. and K.R. All authors contributed to the interpretation of the data, provided critical feedback on paper drafts and approved the final draft.

## Competing interests

The authors declare no competing interests.

## Additional information

**Correspondence and requests for materials** should be addressed to Agnieszka Gidziela or Margherita Malanchini.

**Agnieszka Gidziela** ⑩ [1,2] ✉**, Yasmin I. Ahmadzadeh** ⑩ [2]**, Giorgia Michelini** ⑩ [1,3]**, Andrea G. Allegrini** ⑩ [2,4]**,**
**Jessica Agnew-Blais**[1]**, Lok Yan Lau**[2]**, Megan Duret**[2]**, Francesca Procopio**[2]**, Emily Daly**[2]**, Angelica Ronald** ⑩ [5]**, Kaili Rimfeld**[2,6]
**& Margherita Malanchini** ⑩ [1,2] ✉

[1]School of Biological and Behavioural Sciences, Queen Mary University of London, London, UK. [2]Social, Genetic and Developmental Psychiatry Centre, King's College London, London, UK. [3]UCLA Semel Institute for Neuroscience, Division of Child and Adolescent Psychiatry, University of California, Los Angeles, Los Angeles, CA, USA. [4]Division of Psychology and Language Sciences, University College London, London, UK. [5]Department of Psychological Sciences,  Birkbeck University of London, London, UK. [6]Department of Psychology, Royal Holloway University of London, Egham, UK. ✉e-mail: a.gidziela@qmul.ac.uk; m.malanchini@qmul.ac.uk

# Reporting Summary

## Statistics

For all statistical analyses, confirm that the following items are present in the figure legend, table legend, main text, or Methods section.

| n/a | Confirmed | |
|---|---|---|
| ☐ | ☒ | The exact sample size (*n*) for each experimental group/condition, given as a discrete number and unit of measurement |
| ☐ | ☒ | A statement on whether measurements were taken from distinct samples or whether the same sample was measured repeatedly |
| ☐ | ☒ | The statistical test(s) used AND whether they are one- or two-sided *Only common tests should be described solely by name; describe more complex techniques in the Methods section.* |
| ☐ | ☒ | A description of all covariates tested |
| ☐ | ☒ | A description of any assumptions or corrections, such as tests of normality and adjustment for multiple comparisons |
| ☐ | ☒ | A full description of the statistical parameters including central tendency (e.g. means) or other basic estimates (e.g. regression coefficient) AND variation (e.g. standard deviation) or associated estimates of uncertainty (e.g. confidence intervals) |
| ☐ | ☒ | For null hypothesis testing, the test statistic (e.g. *F*, *t*, *r*) with confidence intervals, effect sizes, degrees of freedom and *P* value noted *Give P values as exact values whenever suitable.* |
| ☒ | ☐ | For Bayesian analysis, information on the choice of priors and Markov chain Monte Carlo settings |
| ☒ | ☐ | For hierarchical and complex designs, identification of the appropriate level for tests and full reporting of outcomes |
| ☐ | ☒ | Estimates of effect sizes (e.g. Cohen's *d*, Pearson's *r*), indicating how they were calculated |

*Our web collection on statistics for biologists contains articles on many of the points above.*

## Software and code

Policy information about availability of computer code

| | |
|---|---|
| Data collection | Publications were searched within the Web of Science, Ovid Medline and Ovid Embase databases. Selection of publications was conducted using the Covidence systematic review software, Veritas Health Innovation, Melbourne, Australia (https://www.covidence.org/). Covidence uses iterative product development processes, and therefore does not use version numbers or years and is a web-based collaboration software platform that streamlines the production of systematic and other literature reviews. |
| Data analysis | Thge data were analyzed and visualized using R version 4.0.4 (R Core Team, 2021) and the following packages: metafor (Viechtbauer & Viechtbauer, 2015), psychometric (Fletcher & Fletcher, 2013), Meta-Analysis with Mean Differences (MAd) (Field & Gillett,2014), meta (Balduzzi, Rücker & Schwarzer, 2019), dmetar (Harrer, Cuijpers, Furukawa & Ebert, 2019) and ggplot2 (Wickham, Chang & Wickham, 2016). <br><br> The code for all analyses is available at https://github.com/CoDEresearchlab/Meta_analysis_NDDs_DICCs <br><br> References: <br> Balduzzi, S., Rücker, G. & Schwarzer, G. How to perform a meta-analysis with R: a practical tutorial. Evid. Based Ment. Health 22, 153–160 (2019). <br> Field, A. P. & Gillett, R. How to do a meta-analysis. Br. J. Math. Stat. Psychol. 63, 665–694 (2010). <br> Fletcher, T. D. & Fletcher, M. T. D. Package 'psychometric'. Recuperado Httpcran Rproject Orgwebpackagespsychometricpsychometric Pdfel 4, (2013). <br> Harrer, M., Cuijpers, P., Furukawa, T. & Ebert, D. D. dmetar: Companion R package for the guide'Doing meta-analysis in R'. R Package Version 00 9000, (2019). <br> R Core Team (2021). R: A language and environment for statistical computing. R Foundation for Statistical Computing, Vienna, Austria. URL https://www.R-project.org/. <br> Viechtbauer, W. & Viechtbauer, M. W. Package 'metafor'. Compr. R Arch. Netw. Package 'metafor' Httpcran R-Proj. Orgwebpackagesmetaformetafor Pdf (2015). <br> Wickham, H., Chang, W., & Wickham, M. H. (2016). Package 'ggplot2'. Create elegant data visualisations using the grammar of graphics. |

For manuscripts utilizing custom algorithms or software that are central to the research but not yet described in published literature, software must be made available to editors and reviewers. We strongly encourage code deposition in a community repository (e.g. GitHub). See our research data policies for further information.

Version, 2(1), 1-189

For manuscripts utilizing custom algorithms or software that are central to the research but not yet described in published literature, software must be made available to editors and reviewers. We strongly encourage code deposition in a community repository (e.g. GitHub). See the Nature Portfolio guidelines for submitting code & software for further information.

# Data

Policy information about availability of data

All manuscripts must include a data availability statement. This statement should provide the following information, where applicable:
- Accession codes, unique identifiers, or web links for publicly available datasets
- A description of any restrictions on data availability
- For clinical datasets or third party data, please ensure that the statement adheres to our policy

The data that support the findings, including the master extraction tables, are available at https://github.com/CoDEresearchlab/Meta_analysis_NDDs_DICCs

Publicly available databases:
Web of science (https://www.webofknowledge.com)
Ovid Medline (https://ovidsp.ovid.com)
Ovid Embase (https://ovidsp.ovid.com)

# Field-specific reporting

Please select the one below that is the best fit for your research. If you are not sure, read the appropriate sections before making your selection.

☐ Life sciences   ☒ Behavioural & social sciences   ☐ Ecological, evolutionary & environmental sciences

For a reference copy of the document with all sections, see nature.com/documents/nr-reporting-summary-flat.pdf

# Behavioural & social sciences study design

All studies must disclose on these points even when the disclosure is negative.

| | |
|---|---|
| Study description | Systematic review and multilevel, random-effects meta-analysis of quantitative behaviour genetics studies reporting on the heritability and environmental influences on neurodevelopmental disorders and/or genetic and environmental correlations between neurodevelopmental disorders and disruptive, impulse control and conduct disorders in childhood and adolescence. |
| Research sample | The studies included were published prior to 4th of May 2021. Studies were only included if 75% or more of the sample consisted of children and/or adolescents. Based on guidelines from the World Health Organization (WHO; https://www.who.int/health-topics/adolescent-health#tab=tab_1), we defined the period from childhood to end of adolescence as ranging from age 4, the earliest age for compulsory schooling, to age 24, the end of adolescence. Studies that had selected participants based on other diagnoses not related to neurodevelopmental disorders and/or disruptive, impulse control and conduct disorder categories or based extreme vulnerability environmental insult unrelated to disorders of interest, such as alcohol abuse, were not included. |
| Sampling strategy | Sample sizes were not predetermined. Estimates derived from included studies were weighted by their standard error during meta-analysis. As a result, studies using larger samples were given more power than studies using smaller samples. |
| Data collection | Searches were conducted across three platforms: Web of Science, Ovid Medline, and Ovid Embase and the outputs managed with the aid of Covidence (https://www.covidence.org/). |
| | After the initial searches were conducted and duplicate studies removed, 8,087 studies met the criteria for the first stage of screening, which involved title and abstract scanning. All titles and abstracts were screened by two independent, blinded reviewers to ensure inter-rater agreement. Conflicts were resolved by a third independent reviewer. After this initial screening phase, 6,834 studies were excluded as deemed not relevant for the purpose of the current meta-analysis. |
| | The title and abstract screening process resulted in a total of 1,253 potentially eligible studies. The full text of each study was screened by two independent, blinded reviewers. Reviewer discrepancies were identified and resolved by a third independent reviewer. This resulted in 289 eligible articles. In addition, during full text screening, relevant review articles, meta-analyses, editorials, and conference abstracts were flagged to aid the potential discovery of further relevant studies by either screening the References sections or contacting the authors of conference abstracts. Through this process 7 additional studies were identified, which resulted in a total of 296 studies included in the current meta-analysis. |
| Timing | Studies were identified during three searches: the primary search conducted on the 20th of January 2021, the secondary search conducted on the 15th of April 2021 and the additional search of other relevant meta-analyses and reviews finalized on the 4th of May 2021. |
| Data exclusions | We excluded multiple-generation family designs (e.g., children-of-twins and in-vitro fertilization) due to the potential confounding in the genetic and environmental estimates that could have resulted from including parental traits in the models decomposing the covariance between family members. Studies that had selected participants based on other diagnoses not related to neurodevelopmental or disruptive, impulse control and conduct disorder categories or based on extreme vulnerability or |

environmental insult unrelated to neurodevelopmental or disruptive, impulse control and conduct disorders, such as alcohol abuse, were also excluded.

Non-participation | This meta-analysis used estimates reported by published studies, for which the participation rate was not recorded.

Randomization | This study was descriptive in nature and no experimental manipulation was involved.

# Reporting for specific materials, systems and methods

We require information from authors about some types of materials, experimental systems and methods used in many studies. Here, indicate whether each material, system or method listed is relevant to your study. If you are not sure if a list item applies to your research, read the appropriate section before selecting a response.

## Materials & experimental systems

| n/a | Involved in the study |
|-----|----------------------|
| ☒ ☐ | Antibodies |
| ☒ ☐ | Eukaryotic cell lines |
| ☒ ☐ | Palaeontology and archaeology |
| ☒ ☐ | Animals and other organisms |
| ☒ ☐ | Human research participants |
| ☒ ☐ | Clinical data |
| ☒ ☐ | Dual use research of concern |

## Methods

| n/a | Involved in the study |
|-----|----------------------|
| ☒ ☐ | ChIP-seq |
| ☒ ☐ | Flow cytometry |
| ☒ ☐ | MRI-based neuroimaging |

