## [Peer review file · Nature Human Behaviour]

Peer Review Information

Journal: Nature Human Behaviour

Manuscript Title: A meta-analysis of genetic effects associated with neurodevelopmental disorders and co-occurring conditions

Corresponding author name(s): Agnieszka Gidziela and Margherita Malanchini

Reviewer Comments & Decisions:

Decision Letter, initial version:

31st May 2022

Dear Ms Gidziela,

Thank you once again for your manuscript, entitled "Genetic influences on neurodevelopmental disorders and their overlap with co-occurring conditions in childhood and adolescence: a meta-analysis.," and for your patience during the peer review process.

Your manuscript has now been evaluated by 3 reviewers, whose comments are included at the end of this letter. Although the reviewers find your work to be of interest, they also raise some important concerns. We are interested in the possibility of publishing your study in Nature Human Behaviour, but would like to consider your response to these concerns in the form of a revised manuscript before we make a decision on publication.

To guide the scope of the revisions, the editors discuss the referee reports in detail within the team, including with the chief editor, with a view to (1) identifying key priorities that should be addressed in revision and (2) overruling referee requests that are deemed beyond the scope of the current study. In the case of your manuscript, we ask that you respond to all reviewer comments in full. Please do not hesitate to get in touch if you would like to discuss the revisions at any point.

Your revised manuscript must comply fully with our editorial policies and formatting requirements. Failure to do so will result in your manuscript being returned to you, which will delay its consideration. To assist you in this process, I have attached a checklist that lists all of our requirements. If you have any questions about any of our policies or formatting, please don't hesitate to contact me.

In sum, we invite you to revise your manuscript taking into account all reviewer and editor comments. We are committed to providing a fair and constructive peer-review process. Do not hesitate to contact us if there are specific requests from the reviewers that you believe are technically impossible or

unlikely to yield a meaningful outcome.

We hope to receive your revised manuscript within two months. I would be grateful if you could contact us as soon as possible if you foresee difficulties with meeting this target resubmission date.

- Include a "Response to the editors and reviewers" document detailing, point-by-point, how you addressed each editor and referee comment. If no action was taken to address a point, you must provide a compelling argument. When formatting this document, please respond to each reviewer comment individually, including the full text of the reviewer comment verbatim followed by your response to the individual point. This response will be used by the editors to evaluate your revision and sent back to the reviewers along with the revised manuscript.
- Highlight all changes made to your manuscript or provide us with a version that tracks changes.

[REDACTED]

We look forward to seeing the revised manuscript and thank you for the opportunity to review your work. Please do not hesitate to contact me if you have any questions or would like to discuss these revisions further.

Sincerely,

Charlotte Payne

Charlotte Payne, PhD
Senior Editor
Nature Human Behaviour

Reviewer expertise:

Reviewer #1: Meta analysis, genetics, neurodevelopment

Reviewer #2: Neurodevelopmental disorders, genetics

Reviewer #3: Neurodevelopmental conditions, genetics

REVIEWER COMMENTS:

Reviewer #1:

Remarks to the Author:

Thank you very much for asking me to review this manuscript. I appreciate the amount of work that went into this manuscript: it is a good source of summative information. Interestingly, there are no surprises (and, therefore, not much novel information/insight). In essence, everything is as anticipated, but it is a good summary that needs to be published. I have no specific comments; it could be published as-is. However, if it is going to be revised, I think it is important to underscore that the overview does not provide much of the novel view but instead reiterates where the field has been for the last few years. Reassuring, but not exciting.

Reviewer #2:

Remarks to the Author:

Gidziela et al. presented a massive meta-analysis on 296 independent studies to investigate the family-based genetic, shared and nonshared environmental influences, as well as SNP-based heritability for NDDs, and the correlations between different NDDs and with DICC. They also assessed those influences on NDDs across sex differences, developmental stages, geographical regions, and population ancestries. Built upon what has been partially reported previously, this study provided a very useful systematic overview of heritability for NDDs, and the correlations between different NDDs and other co-occurring conditions. However, there are still some issues to be addressed, that I hope will be helpful to improve the manuscript. My detailed comments are as below:

Major concerns:

1. For the 296 studies included in the meta-analysis, how many of each are family-based and SNP-based? And how many studies are for each disorder? The 237 family-based studies and 29 SNP-based studies do not add up to 296, am I missing something here? It's a good idea to have Supplementary Tables 28-33 to indicate the study group in each analysis, but I would suggest also preparing an extended master table to list all 296 studies together (in one Excel) and providing additional info such as disorder, sample size, and sex group, etc. Those are basic info but important to help interpret the result.
2. Among the 296 studies, many of which are consecutive studies from the same group in different years, and samples in many studies are from the same cohorts, e.g., TEDS, QNTS, CATSS, etc. It's not quite clear what measures have been taken to remove sample duplicates and avoid double counting? The abstract mentioned samples are over 4 million, how did this total come together?
3. As mentioned, the different number of studies were included in each of the different analyses, which means the number of samples is also different for each analysis, but unfortunately there's no mention of sample size, instead only the number of studies been provided. The number of samples is an important issue to consider for most of the analyses. For one example, when comparing the correlations between NDDs and DICC, the sample size for NDDs (ASD & ADHD) is much larger than DICC (CD and ODD), thus the correlations could be biased due to this unbalanced sample size.
4. When comparing NDDs with DICC, there are only two DICC disorders (CD and ODD) with a limited number of studies to compare with ADHD (6 studies) and ASD (3 studies), this seems to be a weak claim to support the Aim 3 for the overlap with co-occurring conditions, which stands as a main part of the manuscript and in the title and abstract. Why not consider comparing with other more common

co-morbidities with NDDs, such as developmental delay, epilepsy, and other developmental disorders? which also has more studies available.

5. The last two result sections about "Western countries" and "European ancestry" are kind of repetitive to me as they are about more or less the same thing, which is something already well known to the community that NDDs and co-occurring disorders are exclusively studied in Western countries and European ancestry populations. I would suggest integrating them as one section and moving additional content as a supplementary note, this could also help increase the overall readability.

6. I would suggest providing more details in the Methods section on how each of the heritability and correlations values were calculated? If any, I would provide a formula, e.g., what's for I2. Such details will be critical to understanding how the data was processed especially when having multiple studies included in the comparison.

Minor comments:

1. The entire manuscript reads kind of verbose, will be beneficial if can simplify and restructure it a bit. For example, the Introduction section is too long, a large chunk of it is just restate the results from previous studies, which is not necessary to have so much content; Some of the result sections read more like a method description, e.g., lines 170-183, I would move some of this and others into the Methods section.

2. Suggest also adding the group of "NDDs combined" in Figure 3 same as others.

3. Better to keep consistency across all text and figure legends, e.g., h^2 (2 is superscript) was used in the text but h_2 was seen on figures. Correct all others that apply.

4. The saying in line 285 "males are four times more likely to be diagnosed with NDDs" is not true, the ratio of ~4:1 is for ASD but not for all NDDs, both references (52, 53) cited there are also for ASD.

5. Make sure there are enough details in the figure legend to help understand the figures. E.g., in Fig.5 and Fig.S1B, what's the height (density) of the y-axis stand for? This also applies to the supplementary figures.

6. Suggest a better strategy for sub-titles of the result sections.

7. The core input metadata "Extraction_heritability.csv" is not provided in the GitHub, this is needed for others to reproduce the result.

Reviewer #3:

Remarks to the Author:

Key results: This article comprehensively perform meta-analyses on the genetics of neurodevelopmental disorders (NDDs) and their overlaps with each other and with disruptive, impulse control and conduct disorders (DICC). They identify considerable heritability of NDDs and generate new evidence as to why NDDs commonly co-occur.

They also show that the genetic basis of NDDs and DICC have much more in common than previously thought, revealing new directions for future NDD research and potential implications in clinical and educational practice. Importantly, the authors highlight the need to improve diversity of behavioural genetics studies in terms of the conditions studied and the underrepresentation of studies in non-European populations. Finally, the authors examine if genetic effects differed between males and females and found no evidence of this.

Validity: The manuscript is excellent and no major flaws are present.

Originality and significance: The authors have successfully tackled some of the biggest questions in the NDD field in this paper. They have applied systematic review methodology to produce important advances in the field and have advanced the current understanding for why NDDs so frequently co-occur. Perhaps even more exciting is the comprehensive evidence that DICCes also share much of their genetic aetiology. While a systematic review itself is an established research method, the application to this scale of included studies and number of conditions is novel. It represents the first stepping stone of a shift in how we think about NDDs and DICCes that will have clinical and educational importance.

Data & methodology: The authors followed the Preferred Reporting Items for Systematic Reviews and Meta-Analyses (PRISMA) guidelines which is the current best practice for systematic reviews. Code used in the study is publically available. The data is presented extremely clearly, which is commendable – there is a lot of data contained in the supplementary material. The methods are suitable for the research presented and I full confidence could be reproduced.

Preregistration: The study was pre-registered on the international prospective register of systematic reviews (PROSPERO), and no deviations from the method were reported.

Appropriate use of statistics and treatment of uncertainties: Statistical tests were appropriate and were clearly presented.

Custom code: The code used in the article is publically available

Conclusions: The conclusions and data interpretation are robust, valid and reliable.

Suggested improvements: No major changes are required. Please see the minor changes listed below.

References: page 6 line 204 – please provide a reference for the multi-level random effect meta-analysis method used.

Clarity and context: The article is extremely well written and presents the extensive results clearly. My comments are very minor. Abstract – using co-occurrence and co-occurrence with other conditions is a bit confusing in line 17/18 - rephrase. "This meta-analysis bridges our gap – delete and use the words for explaining co-occurrence." – line 18 seems redundant. I don't think the abstract really does the size or scope of this paper justice – rephrase.

Minor comments:

US spelling – etiology, mix of UK/US spelling throughout

Introduction

P2 - Lines 36 – 38 is a bit misleading – can lead to...

The long term outcomes associated with NDDs and DICCes are worth mentioning in the introduction as this adds weight to the justification for understanding their genetic aetiology.

P2 - Line 42 – they can manifest as early as before the child... - rephrase

P2 - Line 69 – I think "what" is missing from this sentence

P3 – line 102 some spacing issues with the genetic correlations reported

P4 – line 139 and 140 – overlaps between ASD and psychopathic traits – both 0.43 and 0.99 are

reported – I think the 0.99 is an error. If it is not an error then this needs explanation.
Figure 1 is a little confusing – the legend does not match. Aim 1 – heritability of NDDs. Aim 2 – it says bottom left but is I think referring to the centre and NDD section. Aim 3 – centre and right section.
Can this be made a bit clearer?

Results

Supplementary table 13 – change spacing so numbers of studies N so more easily visible
P13 line 413 – a bit of a stretch to say mostly in the US as about half were with others in Europe – please reword.

Author Rebuttal to Initial comments

Reviewer 1; comment 1

Thank you very much for asking me to review this manuscript. I appreciate the amount of work that went into this manuscript: it is a good source of summative information. Interestingly, there are no surprises (and, therefore, not much novel information/insight). In essence, everything is as anticipated, but it is a good summary that needs to be published. I have no specific comments; it could be published as-is. However, if it is going to be revised, I think it is important to underscore that the overview does not provide much of the novel view but instead reiterates where the field has been for the last few years. Reassuring, but not exciting.

Response: Thank you for this positive feedback. We revised our discussion to highlight how our meta-analytic results provide a data-driven summary of what has been published in the field.

Reviewer 2; comment 1

Gidziela et al. presented a massive meta-analysis on 296 independent studies to investigate the family-based genetic, shared and nonshared environmental influences, as well as SNP-based heritability for NDDs, and the correlations between different NDDs and with DICCs. They also assessed those influences on NDDs across sex differences, developmental stages, geographical regions, and population ancestries. Built upon what has been partially reported previously, this study provided a very useful systematic overview of heritability for NDDs, and the correlations between different NDDs and other co-occurring conditions.

Response: We are grateful to the reviewer for this positive feedback.

Reviewer 2; comment 2

For the 296 studies included in the meta-analysis, how many of each are family-based and SNP-based? And how many studies are for each disorder? The 237 family-based studies and 29 SNP-based studies do not add up to 296, am I missing something here? It's a good idea to have Supplementary Tables 28-33 to indicate the study group in each analysis, but I would suggest also preparing an extended master table to list all 296 studies together (in one Excel) and providing additional info such as disorder, sample size, and sex group, etc. Those are basic info but important to help interpret the result.

Response: We thank the reviewer for this suggestion.

The number of family-based and SNP-based studies do not add up because some studies provided both family-based and SNP-based estimates. These studies were counted only once towards the grand total but included separately in family-based and SNP-based categories (we now discuss this issue in the revised version of the manuscript, page 5) . We detail the numbers of studies in each category and for each disorder at the start of the Results section (see excerpts below):

‘We identified a total of 236 family-based studies, comprising 2,792,511 partly overlapping individuals, that investigated the proportion of variance in NDDs that is accounted for by genetic factors. Out of the total, 121 studies (N= 682,340) investigated ADHD, 89 studies (N= 360,920) specific learning disorders, 36 studies (N= 1,821,970) ASD, 23 (N= 130,757) studies communication disorders, 6 studies (N= 52,278) motor disorders and 2 studies (N= 9,036) intellectual disabilities.’ [Page 5]

‘Out of the total of 29 SNP-based studies, involving 893,896 partly overlapping individuals, the only disorders that were addressed by at least two independent studies⁴⁹, included ASD (15 studies; N= 637,240), ADHD (14 studies; N= 725,168), specific learning disorders (9 studies; N= 40,637) and communication disorders (4 studies; N= 14,894).’ [Page 5].

Thank you for the excellent suggestion of including a master extraction table. We have now uploaded our two master extraction tables, one table for studies addressing heritability and environmental influences on NDDs and the second for studies addressing genetic & environmental correlations between NDDs and between NDDs and DICCs, to GitHub (we direct our readers to these two master tables on pages 15 and 17 of the revised manuscript). These tables include detailed information on all studies.

Reviewer 2; comment 3

Among the 296 studies, many of which are consecutive studies from the same group in different years, and samples in many studies are from the same cohorts, e.g., TEDS, QNTS, CATSS, etc. It's not quite clear what measures have been taken to remove sample duplicates and avoid double counting? The abstract mentioned samples are over 4 million, how did this total come together?

Response: We thank the reviewer for highlighting the need to clarify this issue. As described in the Methods section (page 16), and in Supplementary Note 9, we ran a multilevel meta-analysis that allowed us to account for the non-independence of estimates derived from partly or completely overlapping samples (i.e., estimates obtained from multiple studies that have used the same cohort of participants) by including cohort as a level. To further account for the non-independence of sampling variance (i.e., when data from partly the same individuals is used to estimate multiple effect sizes), we also aggregated multiple estimates within each individual study (see Methods section page 16, and a detailed description in Supplementary Note 12 and Supplementary Figure 25).

The sample size of 4 million, as we included in the abstract, is the sum of all samples across all studies included in the meta-analysis, and therefore these are not completely independent samples. We state that samples include partly overlapping individuals in the revised manuscript (see Abstract).

Reviewer 2; comment 4

As mentioned, the different number of studies were included in each of the different analyses, which means the number of samples is also different for each analysis, but unfortunately there's no mention of sample size, instead only the number of studies been provided. The number of samples is an important issue to consider for most of the analyses. For one example, when comparing the correlations between NDDs and DICCs, the sample size for NDDs (ASD & ADHD) is much larger than DICCs (CD and ODD), thus the correlations could be biased due to this unbalanced sample size.

Response: We appreciate the reviewer's suggestions, and in the revised manuscript, we include sample size information next to each mention of the number of studies (see for example page 5).

Reviewer 2; comment 5

‘When comparing NDDs with DICC, there are only two DICC disorders (CD and ODD) with a limited number of studies to compare with ADHD (6 studies) and ASD (3 studies), this seems to be a weak claim to support the Aim 3 for the overlap with co-occurring conditions, which stands as a main part of the manuscript and in the title and abstract. Why not consider comparing with other more common co-morbidities with NDDs, such as developmental delay, epilepsy, and other developmental disorders? which also has more studies available.’

Response: We thank the reviewer for this suggestion. When we planned this meta-analysis and preregistered our protocol with the international prospective register of systematic reviews (PROSPERO; https://www.crd.york.ac.uk/prospéro/display_record.php?RecordID=230158), we decided to focus on synthesizing extant literature on the co-occurrence between NDDs and DICC. Our choice was motivated by several reasons. First, both NDDs and DICC onset in childhood and/or adolescence. Second, both NDDs and DICC impact schooling, learning and education but they are rarely considered in conjunction in clinical and educational practice. Third, the lack of available systematic reviews. Because of the very wide scope of this project, we decided to set our boundaries by following the DSM-5 classification, and examining all disorders included under the NDDs and DICC categories. When we preregistered our plan, we did not expect to encounter the major research gap that we uncovered, we highlight this in the Discussion section of the revised manuscript (see excerpt below).

‘The lack of equity in focus across NDDs was pronounced in analyses addressing our third aim. Sources of co-occurrence between NDDs and DICC could only be investigated between ADHD & conduct disorder, ADHD & oppositional defiant disorder and between ASD & conduct disorder. Considering that in the DSM-5 the DICC category comprises 8 distinct disruptive disorders, this highlights a major gap in our knowledge.’ [Page 13]

Reviewer 2; comment 6

The last two result sections about “Western countries” and “European ancestry” are kind of repetitive to me as they are about more or less the same thing, which is something already well known to the community that NDDs and co-occurring disorders are exclusively studied in Western countries and European ancestry populations. I would suggest integrating them as one section and moving additional content as a supplementary note, this could also help increase the overall readability.

Response: We thank the reviewer for this helpful comment. We have integrated the geography and ancestry-related sections into a single section called “Geography and ancestry” in the revised manuscript [Pages 9 and 10].

Reviewer 2; comment 7

‘I would suggest providing more details in the Methods section on how each of the heritability and correlations values were calculated? If any, I would provide a formula, e.g., what’s for I2. Such details will be critical to understanding how the data was processed especially when having multiple studies included in the comparison.’

Response: We appreciate the reviewer’s suggestion and include a detailed description of how variance components and correlations were calculated, including formulas, in the Methods section of the revised manuscript, see excerpt below:

‘Estimates of heritability, shared and nonshared environmental influences were extracted as reported by individual studies. When studies only reported twin correlations, variance components were calculated using the Falconer’s formula, as follows:

$$h^2 = 2(r_{MZ} - r_{DZ})$$

$$c^2 = 1 - (h^2 + e^2)$$

$$e^2 = 1 - r_{MZ}$$

Where: h^2 = family-based heritability; r_{MZ} = monozygotic twin correlation; r_{DZ} = dizygotic twin correlation; c^2 = shared environmental influences; e^2 = nonshared environmental influences.

Genetic, shared and nonshared environmental correlations were only extracted if reported by individual studies.’ [Page 15]

Reviewer 2; comment 8

The entire manuscript reads kind of verbose, will be beneficial if can simplify and restructure it a bit. For example, the Introduction section is too long, a large chunk of it is just restate the results from previous studies, which is not necessary to have so much content; Some of the result sections read more like a method description, e.g., lines 170-183, I would move some of this and others into the Methods section.

Response: We have now shortened and simplified the introduction, which went from 1615 to 933 words. We also, as suggested, moved methodological descriptions from the Results to the Methods section.

Reviewer 2; comment 9

Suggest also adding the group of “NDDs combined” in Figure 3 same as others.

Response: Thank you for this suggestion, we have amended the figure accordingly.

Reviewer 2; comment 10

‘Better to keep consistency across all text and figure legends, e.g., h^2 (2 is superscript) was used in the text but $h2$ was seen on figures. Correct all others that apply.’

Response: We thank the reviewer for this comment. We have now made sure that the format of abbreviations is consistent across text and figures. The only instance where it was challenging to implement this change was heritability. We chose to use h^2 (2 in superscript) in the main text to be consistent with how heritability is conventionally abbreviated in the published literature. However, we found it difficult to add superscript to some of our figures, which were created using R. We have modified the figure legends in line with what displayed in each image.

Reviewer 2; comment 11

The saying in line 285 “males are four times more likely to be diagnosed with NDDs” is not true, the ratio of ~4:1 is for ASD but not for all NDDs, both references (52, 53) cited there are also for ASD.

Response: Thank you for pointing this out. We now specify that this statement refers to ASD.

Reviewer 2; comment 12

Make sure there are enough details in the figure legend to help understand the figures. E.g., in Fig.5 and Fig.S1B, what’s the height (density) of the y-axis stand for? This also applies to the supplementary figures.

Response: We have now changed the figures and legends to include all the necessary details.

Reviewer 2; comment 13

Suggest a better strategy for sub-titles of the result sections.

Response: We have now changed the subtitles. For instance, “While the aetiology of NDDs is comparable for males and females, their co-occurrences differ by sex”, has been changed to: “Sex differences”.

Reviewer 2; comment 14

The core input metadata “Extraction_heritability.csv” is not provided in the GitHub, this is needed for others to reproduce the result.

Response: The master extraction tables “Extraction_heritability.csv” and “Extraction_correlations.csv” are now available on GitHub.

Reviewer 3; comment 1

Key results: This article comprehensively perform meta-analyses on the genetics of neurodevelopmental disorders (NDDs) and their overlaps with each other and with disruptive, impulse control and conduct disorders (DICC)s. They identify considerable heritability of NDDs and generate new evidence as to why NDDs commonly co-occur.

They also show that the genetic basis of NDDs and DICC)s have much more in common than previously thought, revealing new directions for future NDD research and potential implications in clinical and educational practice. Importantly, the authors highlight the need to improve diversity of behavioural genetics studies in terms of the conditions studied and the underrepresentation of studies in non-European populations. Finally, the authors examine if genetic effects differed between males and females and found no evidence of this.

Validity: The manuscript is excellent, and no major flaws are present.

Originality and significance: The authors have successfully tackled some of the biggest questions in the NDD field in this paper. They have applied systematic review methodology to produce important advances in the field and have advanced the current understanding for why NDDs so frequently co-occur. Perhaps even more exciting is the comprehensive evidence that DICC)s also share much of their genetic aetiology. While a systematic review itself is an established research method, the application to this scale of included studies and number of conditions is novel. It

represents the first steppingstone of a shift in how we think about NDDs and DICCs that will have clinical and educational importance.

Data & methodology: The authors followed the Preferred Reporting Items for Systematic Reviews and Meta-Analyses (PRISMA) guidelines which is the current best practice for systematic reviews. Code used in the study is publicly available. The data is presented extremely clearly, which is commendable – there is a lot of data contained in the supplementary material. The methods are suitable for the research presented and I full confidence could be reproduced.

Preregistration: The study was pre-registered on the international prospective register of systematic reviews (PROSPERO), and no deviations from the method were reported.

Appropriate use of statistics and treatment of uncertainties: Statistical tests were appropriate and were clearly presented.

Custom code: The code used in the article is publicly available

Conclusions: The conclusions and data interpretation are robust, valid and reliable.

Suggested improvements: No major changes are required.'

Response: We are very grateful to the reviewer for reading our work so carefully and for the positive feedback.

Reviewer 3; comment 2

References: page 6 line 204 – please provide a reference for the multi-level random effect meta-analysis method used.

Response: We thank the reviewer this comment and we have now added the reference.

Reviewer 3; comment 3

Clarity and context: The article is extremely well written and presents the extensive results clearly. My comments are very minor. Abstract – using co-occurrence and co-occurrence with other conditions is a bit confusing in line 17/18 - rephrase. “This meta-analysis bridges our gap – delete and use the words for explaining co-occurrence.” – line 18 seems redundant. I don't think the abstract really does the size or scope of this paper justice – rephrase.'

Response: We thank the reviewer for this comment. We have now re-structured the abstract according to the reviewer's suggestion so that it reads more clearly and better reflects the scope of this meta-analysis, see excerpt below:

‘We further explored developmental trajectories and the moderating role of gender, measurement, geography, and ancestry.’ [Page 2]

Reviewer 3; comment 4

US spelling – etiology, mix of UK/US spelling throughout.

Response: We thank the reviewer for pointing out this mistake. We have now made the spelling consistent throughout the manuscript.

Reviewer 3; comment 5

‘P2 - Lines 36 – 38 is a bit misleading – can lead to...

The long-term outcomes associated with NDDs and DICCs are worth mentioning in the introduction as this adds weight to the justification for understanding their genetic aetiology.’

Response: We thank the reviewer for this important suggestion. We added description of long-term consequences of NDDs and DICCs to the introduction, see excerpt below:

‘For instance, ADHD in childhood has been associated with an increased risk of educational and occupational problems, risk-taking, and mood disorders in adulthood, and an ASD diagnosis in childhood with increased occupational difficulties and a greater risk of psychopathologies in adulthood. Difficulties are often more salient for those children diagnosed with more than one NDD.’ [Page 3]

And

‘Similar to NDDs, DICCs have been linked to impaired social, emotional, and educational outcomes.’ [Page 4]

Reviewer 3; comment 6

P2 - Line 42 – they can manifest as early as before the child... - rephrase.

Response: We have rephrased.

Reviewer 3; comment 7

‘P2 - Line 69 – I think “what” is missing from this sentence.’

Response: Corrected.

Reviewer 3; comment 8

'P3 – line 102 some spacing issues with the genetic correlations reported.'

Response: Corrected.

Reviewer 3; comment 9

'P4 – line 139 and 140 – overlaps between ASD and psychopathic traits – both 0.43 and 0.99 are reported – I think the 0.99 is an error. If it is not an error, then this needs explanation.'

Response: We thank the reviewer for pointing this out. This indeed is an error as 0.99 referred to shared environmental correlations, rather than genetic correlations. This has now been corrected.

Reviewer 3; comment 10

'Figure 1 is a little confusing – the legend does not match. Aim 1 – heritability of NDDs. Aim 2 – it says bottom left but is I think referring to the centre and NDD section. Aim 3 – centre and right section. Can this be made a bit clearer?'

Response: We thank the reviewer for pointing this out. We have changed the figure to improve clarity.

Reviewer 3; comment 11

'Supplementary table 13 – change spacing so numbers of studies N so more easily visible.'

Response: Changed.

Reviewer 3; comment 12

'P13 line 413 – a bit of a stretch to say mostly in the US as about half were with others in Europe – please reword.'

Response: We thank the reviewer for this suggestion. We have now reworded the sentence.

Decision Letter, first revision:

6th September 2022

Dear Ms Gidziela,

Thank you once again for your manuscript, entitled "A meta-analysis of genetic effects on neurodevelopmental disorders and co-occurring conditions", and for your patience during the peer review process.

Your manuscript has now been evaluated by 2 of our reviewers (Reviewers 2 and 3) from the previous round, and an additional reviewer (Reviewer 4) with meta-analysis expertise. Their comments are included at the end of this letter, and as you will see, all are positive about your revised work, but reviewers 2 and 4 raise some important concerns. We are very interested in the possibility of publishing your study in Nature Human Behaviour, but would like to consider your response to these concerns in the form of a revised manuscript before we make a decision on publication.

To guide the scope of the revisions, the editors discuss the referee reports in detail within the team, including with the chief editor, with a view to (1) identifying key priorities that should be addressed in revision and (2) overruling referee requests that are deemed beyond the scope of the current study. We hope that you will find the prioritised set of referee points to be useful when revising your study. Please do not hesitate to get in touch if you would like to discuss these issues further.

1) Please address in full all of the requests made by Reviewer 4, including running outlier case identification and influential case identification analyses, and presenting these results in the manuscript.

2) In their comments, Reviewer 4 asks that you follow the PRISMA 2020 extension for Abstracts. We ask that you provide all the information required by the PRISMA abstract checklist, but please do not use the structured format. We would be able to extend the word limit for your Abstract up to 250 words to allow for this information to be included, if necessary.

3) We would also like to request changes to Figure 7. Firstly, we recommend moving Panel C to your Supplementary Information, as this shows information for only three countries, and its removal would allow more space for the other panels. While Springer Nature is neutral with respect to territorial disputes and the choice of specific maps rests with you, we ask that you use the same standard map across panels, and we do ask you to take into consideration Reviewer 2's other comments. When amending this figure please also replace the red and green colours as they are not suitable for colourblind readers, and please remove the eurocentric circles from all maps.

4) Please provide an updated PRISMA checklist when submitting your revised manuscript, and please provide in the main text all the required information, as requested by Reviewer 4.

In sum, we invite you to revise your manuscript taking into account all reviewer and editor comments. We are committed to providing a fair and constructive peer-review process. Do not hesitate to contact us if there are specific requests from the reviewers that you believe are technically impossible or unlikely to yield a meaningful outcome.

We hope to receive your revised manuscript within two months. I would be grateful if you could contact us as soon as possible if you foresee difficulties with meeting this target resubmission date.

- Include a "Response to the editors and reviewers" document detailing, point-by-point, how you addressed each editor and referee comment. If no action was taken to address a point, you must provide a compelling argument. When formatting this document, please respond to each reviewer comment individually, including the full text of the reviewer comment verbatim followed by your response to the individual point. This response will be used by the editors to evaluate your revision and sent back to the reviewers along with the revised manuscript.
- Highlight all changes made to your manuscript or provide us with a version that tracks changes.

[REDACTED]

We look forward to seeing the revised manuscript and thank you for the opportunity to review your work. Please do not hesitate to contact me if you have any questions or would like to discuss these revisions further.

Sincerely,

Charlotte Payne

Charlotte Payne, PhD
Senior Editor
Nature Human Behaviour

REVIEWER COMMENTS:

Reviewer #2:
Remarks to the Author:

The authors have addressed most of my comments satisfactorily. I only have two minor comments left:

1. For my comment 4, the authors only added the sample size but no commentary on the correlations could be biased due to the unbalanced sample size. I would suggest adding one sentence or two to discuss this possible bias and draw a soft conclusion about the correlation.
2. The world maps in Figure 7 are not consistent across all three panels, panel A is even different from B and C; and it's incomplete for some regions, e.g., the Hainan and Taiwan islands of China are not always displayed; and the coloring for countries could be controversial. This figure needs to be fixed.

Reviewer #3:
Remarks to the Author:

My comments have been fully addressed. Congratulations on producing an excellent and important paper.

Reviewer #4:
Remarks to the Author:

Thank you for the opportunity to review this stellar manuscript. It reports a massive meta-analysis on the aetiology of neurodevelopmental disorders (NDDs). Although -by definition- meta-analyses do not communicate novel findings, they are nevertheless extremely valuable, transparent, and systematic summaries of available evidence. I am looking forward to seeing this meta-analysis take its place in the published literature and help shape future research and practice. I only have a few comments to make, which are directed towards a closer adherence to the PRISMA 2020 guidelines as well as to achieving completeness in the analysis.

1. Abstract.

The abstract ideally needs to follow the PRISMA 2020 extension for Abstracts. A checklist is to be found here: https://prisma-statement.org/documents/PRISMA_2020_abstract_checklist.pdf. At the moment, a number of elements are missing from the abstract, for example inclusion and exclusion criteria, information sources, whether a risk of bias analysis was conducted, and summary estimates and confidence intervals. I understand that there is a strict word limit for the abstract, but maybe the authors can fit in some of this info. They are also advised to submit the PRISMA 2020 extension for Abstracts checklist as part of the supplementary material.

2. PRISMA 2020

I would also ask the authors to submit the PRISMA 2020 checklist as part of the supplementary material (https://prisma-statement.org/documents/PRISMA_2020_checklist.pdf). This will help the reader identify which elements of PRISMA 2020 are currently missing, but it will also help the authors add those elements, where possible, to their manuscript. Some elements that are currently missing are (the following list is not aiming at being exhaustive):

- a. A section describing deviations from the preregistration (it could be placed in the supplementary

material)

b. A risk of bias analysis. This analysis is also oftentimes called "quality assessment", as is the case in the present manuscript. The authors do provide a list with the criteria they used to conduct this analysis in the supplementary material (supplementary note 10), but they do not report on the findings of this assessment (e.g., in tabular format or using a traffic-lights figure). Moreover, did they repeat any analyses in case the quality of the included studies was low, after removing these low quality studies as a form of sensitivity analysis?

c. Inter-rater agreement: it is reported that this was calculated both for study screening and quality assessment, but the actual numerical value of the inter-rater agreement is missing, Moreover, was inter-rater agreement calculated for data extraction too?

d. Certainty assessment

e. Discussion of limitations of the review process used

3. Extra analyses

This meta-analysis is rich in information, but for the sake of completeness I would ask the authors to further run an outlier cases identification and influential cases identification (there is available R code for this, see here:

https://bookdown.org/MathiasHarrer/Doing_Meta_Analysis_in_R/qanda.html#qanda2, so it should be pretty straight-forward).

Author Rebuttal, first revision:

Reviewer 2; comment 1

“The authors have addressed most of my comments satisfactorily. I only have two minor comments left:

1. For my comment 4, the authors only added the sample size but no commentary on the correlations could be biased due to the unbalanced sample size. I would suggest adding one sentence or two to discuss this possible bias and draw a soft conclusion about the correlation.”

Response: We thank the Reviewer for this suggestion. We have now added several sentences commenting on the sample size imbalance (see excerpts below):

‘However, given the considerable differences in sample size used to derive genetic correlations between pairs of disorders, for example between ASD & ADHD or communication disorders & motor disorders, the strength of these correlations may be difficult to compare. Low correlations could also reflect low power to detect the true overlap.’ [Page 6].

‘The similar extent of genetic overlap between ADHD & conduct disorder or ADHD & oppositional defiant disorder and ADHD & ASD may not be free from biases introduced by an

unbalanced sample size used to derive these meta-analytic estimates. In addition, large meta-analytic standard errors make assessing the significance of differences between the estimates difficult.’ [Page 7].

Reviewer 2; comment 2

“2. The world maps in Figure 7 are not consistent across all three panels, panel A is even different from B and C; and it’s incomplete for some regions, e.g., the Hainan and Taiwan islands of China are not always displayed; and the coloring for countries could be controversial. This figure needs to be fixed.”

Response: We thank the Reviewer for this comment. We have now standardized the maps and changed the colour schemes.

Reviewer 3; comment 1

‘My comments have been fully addressed. Congratulations on producing an excellent and important paper.’

Response: We are grateful for this positive feedback.

Reviewer 4; comment 1

‘Thank you for the opportunity to review this stellar manuscript. It reports a massive meta-analysis on the aetiology of neurodevelopmental disorders (NDDs). Although -by definition- meta-analyses do not communicate novel findings, they are nevertheless extremely valuable, transparent, and systematic summaries of available evidence. I am looking forward to seeing this meta-analysis take its place in the published literature and help shape future research and practice. I only have a few comments to make, which are directed towards a closer adherence to the PRISMA 2020 guidelines as well as to achieving completeness in the analysis.’

Response: We thank the Reviewer for agreeing to review our manuscript and we appreciate the positive feedback and the excellent suggestions.

Reviewer 4; comment 2

‘1. Abstract.

The abstract ideally needs to follow the PRISMA 2020 extension for Abstracts. A checklist is to be found here: https://prisma-statement.org/documents/PRISMA_2020_abstract_checklist.pdf. At the moment, a number of

elements are missing from the abstract, for example inclusion and exclusion criteria, information sources, whether a risk of bias analysis was conducted, and summary estimates and confidence intervals. I understand that there is a strict word limit for the abstract, but maybe the authors can fit in some of this info. They are also advised to submit the PRISMA 2020 extension for Abstracts checklist as part of the supplementary material.'

Response: We thank the Reviewer for this suggestion. The revised Abstract includes the requested information. We have also added the PRISMA 2020 extension for Abstracts checklist to the Supplementary Material (see Supplementary Note 10).

Reviewer 4; comment 3

'2. PRISMA 2020

I would also ask the authors to submit the PRISMA 2020 checklist as part of the supplementary material (https://prisma-statement.org/documents/PRISMA_2020_checklist.pdf). This will help the reader identify which elements of PRISMA 2020 are currently missing, but it will also help the authors add those elements, where possible, to their manuscript.'

Response: We have added the PRISMA 2020 checklist to the Supplementary Material (see Supplementary Note 9).

Reviewer 4; comment 4a

'Some elements that are currently missing are (the following list is not aiming at being exhaustive):

a. A section describing deviations from the preregistration (it could be placed in the supplementary material)'

Response: We now describe deviations from the PROSPERO-registered protocol in Supplementary Note 8. Deviations included changes in: confirmatory search terms, screening recording method, and risk of bias assessment.

Reviewer 4; comment 4b

'b. A risk of bias analysis. This analysis is also oftentimes called "quality assessment", as is the case in the present manuscript. The authors do provide a list with the criteria they used to conduct this analysis in the supplementary material (supplementary note 10), but they do not report on the findings of this assessment (e.g., in tabular format or using a traffic-lights figure).

Moreover, did they repeat any analyses in case the quality of the included studies was low, after removing these low quality studies as a form of sensitivity analysis?’

Response: We thank the Reviewer for this comment. All extracted papers (296) were assessed for publication (reporting) bias (Supplementary Note 7, Supplementary Tables 13-15, and Supplementary Figures 8-24). In the revised manuscript we report our analyses of the study quality assessment, which we conducted for the first 82 papers that were extracted. As shown in the newly created traffic light plot (Supplementary Figure 25), none of the studies showed a high risk of bias in any of the quality assessment domains considered. In fact, 93.8% of studies showed a low risk of bias across all 9 quality checklist items, and the remaining 6.2% showed moderate risk. As a consequence, given the generally low bias, we did not repeat the analyses excluding low-quality studies.

Reviewer 4; comment 4c

‘c. Inter-rater agreement: it is reported that this was calculated both for study screening and quality assessment, but the actual numerical value of the inter-rater agreement is missing. Moreover, was inter-rater agreement calculated for data extraction too?’

Response: We thank the Reviewer for this comment. We calculated rates of inter-rater reliability as the proportion of conflicts to the total number of studies screened and we have added this additional information to Figure 2. Raters were in agreement for 85% of the title and abstract screenings and 76% of the full-text screenings. Inter-rater agreement was not calculated for data extraction. Instead, a secondary reviewer double-checked a randomly selected 10% of all extracted studies. The primary reviewer discussed extraction-related issues and doubts with a secondary reviewer, as per our preregistered protocol.

Reviewer 4; comment 4d

‘d. Certainty assessment’

Response: We now provide a written evaluation of our confidence in the body of evidence collected in Supplementary Note 16 (see excerpt below):

‘We evaluated our confidence in the body of research included in the present meta-analysis based on a number of key factors: (a) the sample size of each study, (b) the consistency of findings across studies, (c) study quality and risk of publication bias.

- (a) Because differences in sample size can introduce an imbalance in the power to estimate effects reliably across studies, in our meta-analysis we weighted each estimate by the standard errors. Estimates reported by studies conducted in larger samples had smaller standard errors and were therefore given more weight if compared to studies conducted in smaller samples.
- (b) The consistency of findings across studies was assessed by visually examining forest plots. Overall, we did not find significant differences between estimates.
- (c) Study quality and risk of bias were assessed in line with the framework proposed by Kmet, Cook and Lee (2004) (see Supplementary Note 13 and Supplementary Figure 25). We applied Egger's regression and inspected funnel plots to examine the impact of publication bias on our results, the outcomes of these analyses are reported in Supplementary Note 7 and Supplementary Tables 13-15 and Supplementary Figures 8-24.

Based on these criteria, we place confidence in the results of the current meta-analysis that shows that: 1) NDDs in childhood and adolescence are highly heritable; 2) that the pattern of co-occurrence between NDDs is complex, and while some NDDs are closely related, others show little genetic overlap; and 3) NDDs show a moderate-to-strong genetic overlap with DICC's.'

Reviewer 4; comment 4e

'e. Discussion of limitations of the review process used'

Response: We have added a discussion of the potential limitations that applied to the review process and the design of the current meta-analysis (see Supplementary Note 17 and excerpt below).

“The review process of the current meta-analysis does not come without limitations. A first limitations is our sole focus on childhood and adolescence. A second limitation relates to our choice of focusing on specific co-occurring conditions, DICC's, without considering other neurological disorders that have been found to co-occur with NDDs, such as epilepsy, cerebral palsy, sleep, or psychiatric disorders. The inclusion of a wider range of co-occurring conditions could have resulted in a more detailed characterization of aetiological overlaps between NDDs and other conditions.

A third limitation is that the current meta-analysis only focused on single-generation studies, i.e., twin and sibling studies and excluded multi-generational family designs, such as

children-of-twins and in-vitro fertilization studies. Future studies focusing on multi-generational designs could provide valuable insights into the role that parental genotypes and correlated environmental influences play in offspring's NDDs and their co-occurring conditions.”

Reviewer 4; comment 5

‘3. Extra analyses

This meta-analysis is rich in information, but for the sake of completeness I would ask the authors to further run an outlier cases identification and influential cases identification (there is available R code for this, see here: https://bookdown.org/MathiasHarrer/Doing_Meta_Analysis_in_R/qanda.html#qanda2, so it should be pretty straight-forward).’

Response: We have now run the suggested analyses. Results of the outlier cases identification analysis are presented in Supplementary Tables 10-12 and results of the influential cases identification analysis are presented in Supplementary Figures 5-7.

Decision Letter, second revision:

Our ref: NATHUMBEHAV-22030550C

21st November 2022

Dear Dr. Gidziela,

Thank you for submitting your revised manuscript "A meta-analysis of genetic effects on neurodevelopmental disorders and co-occurring conditions." (NATHUMBEHAV-22030550C). It has now been read by Reviewer 4 from the previous round, and their comments are below. As you can see, the reviewer finds that the paper has improved in revision. We will therefore be happy in principle to publish it in Nature Human Behaviour, pending minor revisions to comply with our editorial and formatting guidelines.

We are now performing detailed checks on your paper and will send you a checklist detailing our editorial and formatting requirements within a week. Please do not upload the final materials and make any revisions until you receive this additional information from us.

Sincerely,

Charlotte Payne

Charlotte Payne, PhD
Senior Editor
Nature Human Behaviour

Reviewer #4 (Remarks to the Author):

All my comments have been addressed to my satisfaction. I congratulate the authors for they excellent work!

Final Decision Letter:

Dear Ms Gidziela,

We are pleased to inform you that your Article "A meta-analysis of genetic effects associated with neurodevelopmental disorders and co-occurring conditions.", has now been accepted for publication in Nature Human Behaviour.

Please note that *Nature Human Behaviour* is a Transformative Journal (TJ). Authors whose manuscript was submitted on or after January 1st, 2021, may publish their research with us through the traditional subscription access route or make their paper immediately open access through payment of an article-processing charge (APC). Authors will not be required to make a final decision about access to their article until it has been accepted. IMPORTANT NOTE: Articles submitted before January 1st, 2021, are not eligible for Open Access publication. Find out more about Transformative Journals

With best regards,

Charlotte Payne

Charlotte Payne, PhD
Senior Editor
Nature Human Behaviour